# Muscles in Time: Learning to Understand Human Motion by Simulating Muscle Activations

David Schneider*†    Simon Reiß†    Marco Kugler†    Alexander Jaus†    Kunyu Peng†

Susanne Sutschet†    M. Saquib Sarfraz‡    Sven Matthiesen†    Rainer Stiefelhagen†

## Abstract

Exploring the intricate dynamics between muscular and skeletal structures is pivotal for understanding human motion. This domain presents substantial challenges, primarily attributed to the intensive resources required for acquiring ground truth muscle activation data, resulting in a scarcity of datasets. In this work, we address this issue by establishing *Muscles in Time (MinT)*, a large-scale synthetic muscle activation dataset. For the creation of *MinT*, we enriched existing motion capture datasets by incorporating muscle activation simulations derived from biomechanical human body models using the OpenSim platform, a common approach in biomechanics and human motion research. Starting from simple pose sequences, our pipeline enables us to extract detailed information about the timing of muscle activations within the human musculoskeletal system. *Muscles in Time* contains over nine hours of simulation data covering 227 subjects and 402 simulated muscle strands. We demonstrate the utility of this dataset by presenting results on neural network-based muscle activation estimation from human pose sequences with two different sequence-to-sequence architectures.
Data and code are provided under https://simplexsigil.github.io/mint.

## 1   Introduction

Like prisoners in Plato's cave, neural networks for human motion understanding often rely on indirect representations rather than direct, biologically grounded data. In Plato's allegory, prisoners in a cave see only shadows cast on the wall, not the true objects. Similarly, neural networks trained on accessible data, such as RGB and depth-based video recordings or motion capture, only perceive surface-level appearance of motion in contrast to the inner mechanics of the human body.

This reliance on external visual observations provides an incomplete understanding of the true complexities of human motion. Just as the prisoners lack a direct view of the objects casting the shadows, current models lack exposure to the internal workings of the human body, such as the muscle activations driving motion. This gap limits their ability to develop an in-depth understanding of physical exertion, motion difficulty, and mass impact on the body.

Our community has progressed from capturing human motion with camera sensors and predicting activities to pose-based recognition systems that account for the body and its motion over time. These advances, while significant, still overlook the interplay of muscle activations, which are the root of pose sequences and patterns.

---

*Corresponding author: david.schneider@kit.edu
†Karlsruhe Institute of Technology
‡Mercedes-Benz Tech Innovation

38th Conference on Neural Information Processing Systems (NeurIPS 2024) Track on Datasets and Benchmarks.

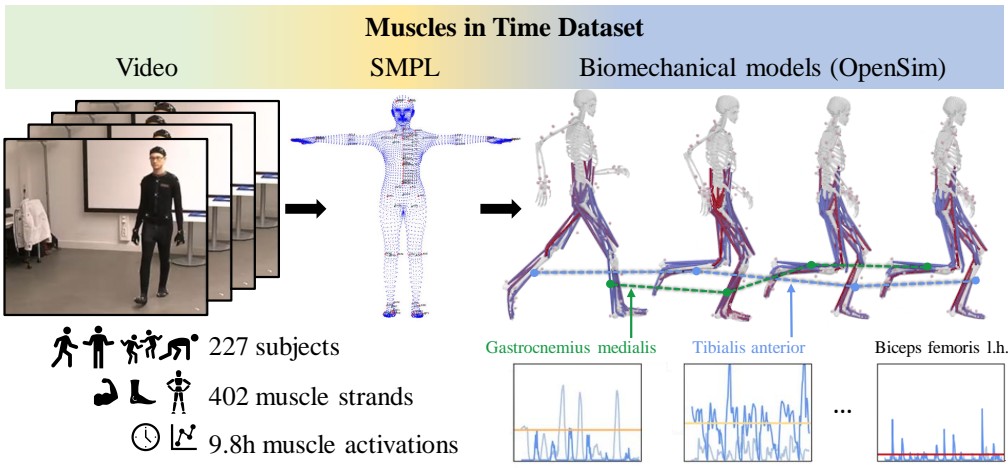

Figure 1: Simulation pipeline of the Muscles in Time dataset. The SMPL representation is extracted from videos, then, the SMPL represented motions are mapped to bio-mechanically validated human body models to simulate fine-grained muscle activation, connecting computer vision with biomechanical research. Bottom right: two activation sequences for exemplary muscles. Images from [47, 15] .

Collecting electromyographic (EMG) data or more commonly used surface electromyographic (sEMG) data, as a measure of muscle activation, presents challenges. It is resource intensive, requiring specialized equipment, controlled environments, and is an invasive procedure. Existing EMG and sEMG datasets are small, limited in scope, and not representative of the variety of human motions. These limitations hinder the development of neural networks that can generalize across different types of motion and subjects.

While acknowledging the contributions of EMG and sEMG datasets, we identify an opportunity to supplement this domain with a synthetic dataset that overcomes some limitations of real-world data collection. The strength of our dataset lies in its scale and detail of muscle activation data, a feat not achievable through conventional methods alone.

Every dataset, simulated or real, has domain-specific fidelity and relevance. Real-world recordings offer authenticity that underpins our understanding of human biomechanics with nuances, such as EMG measurements being subject-specific and varying over the course of one day. Simulated datasets, like ours, offer a complementary perspective by providing comprehensive data for the understanding of muscle activation patterns through a scalable data acquisition pipeline.

In this work, we present a comprehensive large-scale dataset incorporating muscle activation information. We enrich existing motion capture datasets with muscle activation simulations from biomechanical models of the human body. Our pipeline uses simple pose and shape sequences with estimated weight and mass of the human body to simulate muscle activations for individual movements. Using this, we generate the muscles' activation that fit the provided human motions. Figure 1 provides an overview of our pipeline.

We showcase the utility of muscle activations as an additional data type for human motion understanding and gather insights by visualizing the intricate details of our data. Our dataset, the first of its magnitude and detail, describes muscle activation across a wide array of movements. By enhancing the current set of tools available to researchers, we expand the potential for scientific investigation and innovation in the study of human motion.

## 2 Related Work

**Human Motion Analysis and Datasets** EMG-based muscle activation analysis is a well-established field in biomechanical research. Still, publicly available databases including experimentally measured muscle excitation using sEMG are often small in size or cover a small range of muscles or motion variations [21, 30, 79, 25, 51, 46, 61, 42, 34, 50, 65]. The dataset proposed by Zhang *et al.* [79]

Table 1: Comparison between recent muscle activation datasets and *Muscles in Time*.

| | Year | Subjects | Act. Vals. | Duration | Actions | GRF | RGB | Depth | Activation | Skeleton | Description |
|---|---|---|---|---|---|---|---|---|---|---|---|
| **Camargo *et al.* [5]** | 2021 | 22 | 11 | 10 min | 4 | × | × | × | ✓ | × | × |
| **Feldotto *et al.* [21]** | 2022 | 5 | 7 | 10 min | 4 | × | × | × | ✓ | × | × |
| **KIMHu [30]** | 2023 | 20 | 4 | 10 h | 3 | ✓ | ✓ | ✓ | ✓ | ✓ | × |
| **MuscleMap [55]** | 2023 | N/A | 20[a] | ~25 h[b] | 135 | ✓ | ✓ | ✓ | × | ✓ | × |
| **MiA [10]** | 2023 | 10 | 8 | 12.5 h | 15 | ✓ | ✓ | × | × | ✓ | × |
| **MinT (ours)** | 2024 | 227 | 402[c,d] | 10 h | 187 | ✓[d] | ✓[d,e] | ✓[d] | ✓[d] | ✓ | ✓ |

[a] Clip-wise binary labels.   [b] Coarse estimation based on 15,004 clips of 3-9s.   [c] Muscle strands, some muscles represented by multiple strands.   [d] Simulated data.   [e] From [64]

contains 5 persons and leveraged 8 EMG sensors. The KIMHu dataset [30], for example, includes sEMG data of four upper limb muscles measured during different arm exercises performed by 20 subjects. The MIA Dataset [10] includes sEMG signals for eight muscles in total (upper and lower limb) across 10 subjects who performed 15 different exercises, e.g., running, jumping jacks, squats, and elbow punches. MuscleMap is a video-based muscle activation estimation dataset, which assigns binary muscle activation labels to action categories, involving 20 muscle groups and 135 actions [55]. In Table 1, we provide a comparison of multiple recent EMG datasets to MinT. Most notably MinT features a significantly larger number of subjects, a larger number of activation measurements and a diverse range of motions.

OpenSim is an open-source software platform for musculoskeletal modeling, simulation, and analysis. It is widely used in various research areas such as biomechanics research, orthopedics and rehabilitation science, and medical device design [16, 66]. The state-of-the-art process in OpenSim for simulating muscle activations of a certain task requires subject-specific motion and force data. In most cases, these data are obtained through experimental studies, which can be time-consuming and resource-intensive.

In a related field, musculoskeletal humanoid control and simulation focuses on developing computational models and control strategies for simulating human motion with musculoskeletal detail. Recent work by Jiang et al. [35], Caggiano et al. [4], Feng et al. [23], and He et al. [83, 29] has advanced methods for efficient and realistic simulation of muscle-actuated characters. While these approaches differ from OpenSim's focus, they highlight the broader interest in understanding and simulating human musculoskeletal dynamics.

**Skeleton-based Vision Models** Skeleton-based action recognition [22, 1] is pivotal in decoding human actions from video footage, providing a streamlined and insightful depiction of human poses and movements that remains invariant to changes in appearance, illumination, and backdrop. This approach enhances the identification of dynamic skeletal characteristics essential for precise action recognition, finding utility across surveillance, human-computer interaction, and medical fields. The goal of skeleton-based action recognition is to classify actions based on skeletal geometry information [36, 44, 49, 19, 56, 74, 54, 72, 7]. Predominantly, the techniques employed are based on graph convolutional neural networks (GCN)[38, 76, 68, 9, 77, 8], with newer methods adopting transformer architectures [69, 58, 41, 81, 17, 73]. Chen *et al.* [8] proposed channel-wise topology refinement graph convolution for skeleton-based action recognition. Yan *et al.* [75] proposed skeleton masked auto encoder to achieve skeleton sequence pretraining which delivers promising benefits for the skeleton based action recognition. Apart from the GCN and transformer based models, PoseC3D is proposed by Duan *et al.* [19] to use 3D convolutional neural networks on the heat map figures painted by the skeleton joints.

**Sequence-to-sequence Models** Sequence to sequence models [52, 37, 11, 80, 43, 67, 24] are a class of deep neural network architectures designed to transform sequences from one domain into sequences in another domain, typically used in applications such as machine translation, speech recognition, and text summarization. These models generally consist of an encoder that processes the input sequence and a decoder that generates the output sequence, facilitating the learning of complex

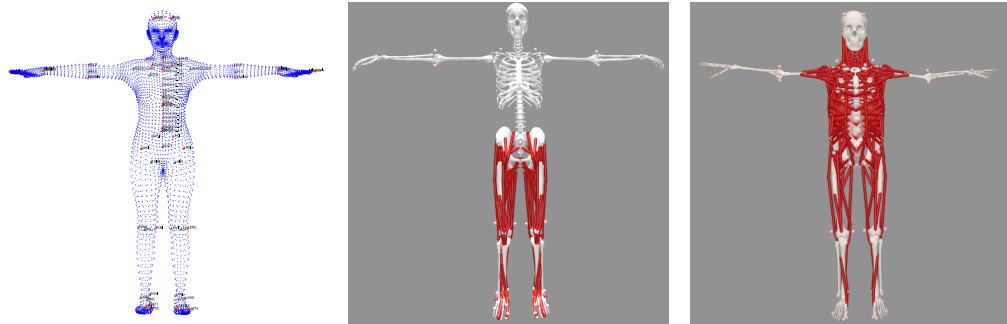

Figure 2: The AMASS body model with specific indices mapped onto the OpenSim lower body model by Lai *et al*. [40] (middle) and model of the thoracolumbar region by Bruno *et al*. [3] (right). Best viewed by zooming in.

sequence mappings through recurrent neural networks (RNNs) [45, 53, 57, 33] or transformer-based architectures [18, 32]. Chan *et al*. [6] proposed Imputer method by using imputation and dynamic programming to achieve sequence modelling. Colombo *et al*. [13] used guiding attention for sequence-to-sequence modelling for dialogue activities prediction. Rae *et al*. [60] proposed compressed transformer architecture for long-range sequence modelling. Foo *et al*. [24] proposed a unified pose sequence modelling method for human behavior understanding.

## 3   The Muscles In Time dataset

To develop the *Muscles in Time (MinT)* framework, we harnessed the comprehensive AMASS dataset, which consolidates various marker-based motion capture (mocap) sequences into a uniform representation using the MoSh++ method, resulting in Skinned Multi-Person Linear Model (SMPL) parametric representations for body pose and shape. AMASS amalgamates mocap data from multiple sources, including the KIT Whole-Body Human Motion Database [48], BMLrub, and BMLmovi [27], encompassing over 11,000 motion captures from more than 300 subjects. This extensive collection enables the analysis of a broad array of human movements, providing a rich basis for studying diverse motion patterns.

The SMPL model serves as a pivotal link, translating mocap data from AMASS into mesh representations which we use to transfer the data into a format compatible with the OpenSim [15] platform. OpenSim is instrumental in constructing intricate biomechanical models that simulate the musculoskeletal system's physical and mechanical properties, allowing for an in-depth analysis of human motion. These models are intricate, requiring precise definitions of joints, masses, inertia, and muscle parameters, such as maximum isometric force, which act as the force-generating actuators.

In this work, we abstain from developing new biomechanical models due to the complexity and expertise required. Instead, we utilize established, pre-validated models, specifically the lower body model by Lai *et al*. [40] and the thoracolumbar region body model by Bruno *et al*. [3], see Figure 2. These models simulate muscle activations for an extensive network of individual muscle strands across various muscle groups, providing a comprehensive simulation of human musculature. A detailed list of these muscle groups and their function in the human body is provided in the Appendix.

Tailoring body model parameters to an individual's anatomical properties results in similar difficulties as with the creation of new body models, therefore parameters are commonly used as specified in the validated original models [40, 3], in the OpenSim community. We follow this approach, providing simulation results for standard models rather than subject-specific human bodies.

To integrate human motion data from AMASS with OpenSim, we map virtual mocap markers to the SMPL-H body mesh's surface vertices, following the method proposed by Bittner *et al*. [2]. This results in a selection of 67 strategically placed vertices that represent marker positions on the body mesh, visualized on the left of Figure 2. We deliberately exclude soft tissue dynamics from the SMPL-H mesh generation to maintain consistent marker positions during motion.

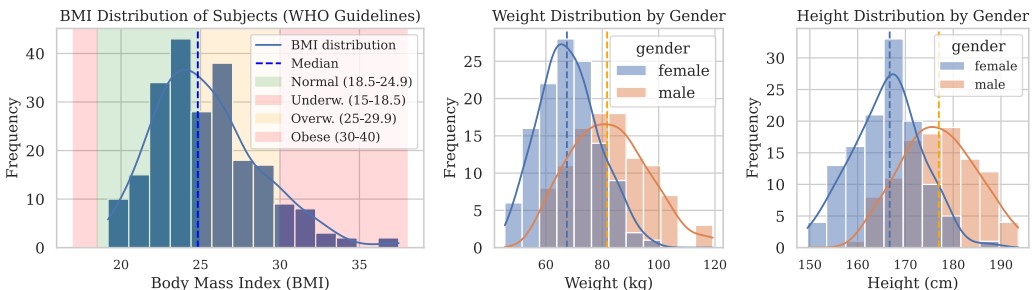

Figure 3: Approximated weight and height distribution of the analysed subjects in the MinT dataset.

Despite OpenSim's automatic scaling capabilities, manual adjustments of marker positions are sometimes necessary to reconcile differences between simulated and real-world data. These adjustments are made on a subject-specific basis, rendering our pipeline semi-automatic. The manually adjusted marker positions are documented and shared to ensure the reproducibility of our simulations.

AMASS lacks data on external ground reaction forces or contact forces, which are crucial for realistic motion simulation. To address this, we integrate the OpenSimAD [20] implementation used in the OpenCap [70] project, which calculates ground reaction forces based on kinematic data and the musculoskeletal model. We employ a tailored parameter setup to optimize the trajectory problem, balancing computational load and accuracy.

Kinematic data is analyzed using OpenSim's *Inverse Kinematics* method. Muscle activations for the lower body are derived from a trajectory optimization problem described in [70]. The estimated ground reaction forces from this problem serve as inputs for the *Static Optimization* method, which calculates muscle activations for the thoracolumbar region.

Due to the computational demands of the trajectory optimization problem, we process the data in segments, ensuring manageable computation times without compromising the continuity of the motion capture sequences. We implement overlapping buffers to mitigate inaccuracies during segment processing, discarding data that fails to meet our stringent error tolerance criteria to maintain a high standard of data quality. Further details on implementation and design decisions of our simulation process are presented in the Appendix.

The Muscles in Time (MinT) dataset represents a significant contribution to the field of biomechanical and computer vision simulation. By integrating and refining existing methodologies, we present a robust pipeline that facilitates the accurate simulation of human muscles in motion by combining established biomechanical models with high quality mocap data. To ensure reproducibility, we will release all relevant data and details of our simulation process to the scientific community.

## 3.1 Dataset Composition

Due to missing information on external forces based on object interactions, inaccurate motion capture recordings or non-converging simulations, the *MinT* dataset covers a subset of its originating datasets in AMASS and does not follow their respective dataset statistics.

**Anthropometrics** While the motion capture recordings in AMASS provide gender labels, information about subjects height and weight is approximated from the SMPL body model. Body weight is calculated by volume resulting from average shape parameters, which follows the approach of Bittner *et al*. [2]. The weight is relevant for the calculation of ground reaction forces and the distribution of weight in the model, affecting the muscle activation in different parts of the body.

The Figure 3 shows the distribution of weight, indicating significant diversity. Underweight subjects are slightly underrepresented in the dataset, subjects in the obese range are well represented.

**Composition of Subdatasets** Within AMASS, *MinT* is limited to the subdatasets EyesJapan, BMLrub, KIT, BMLmovi, and TotalCapture. Figure 9 in the appendix shows the ratio of the originating subdatasets in our final simulation results as well as the average sequence length within these subdatasets. The short sequences in BMLmovi typically depict single activities, while the longer

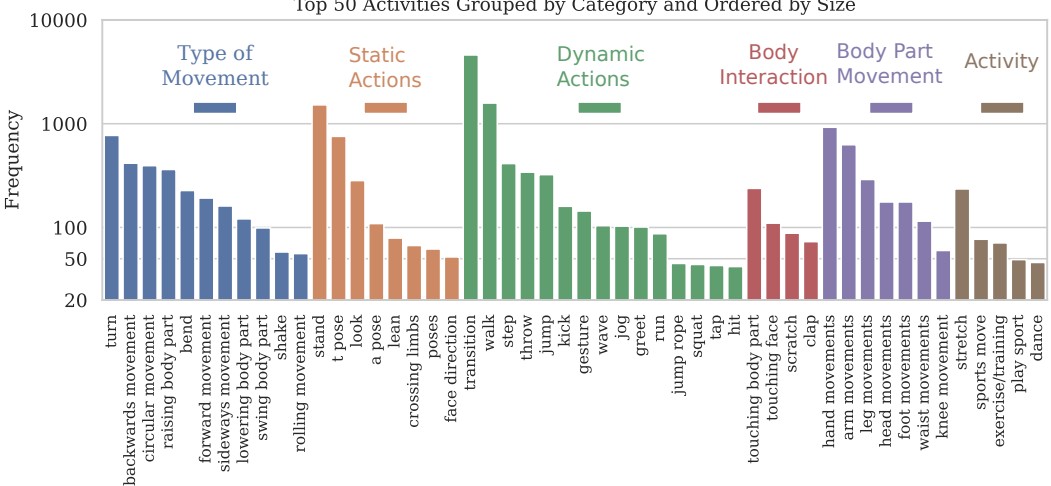

Figure 4: Prevalence of different motions in the MinT dataset.

ones for example in JapanEyes capture a more diverse range of motions within a single sequence. Since we compute activation information for shorter segments and rejoin them afterwards, longer sequences are more prone to gaps in the analysis due to individually failing segment computations.

**Motion Diversity** Figure 4 displays the frequencies of grouped activities on a logarithmic scale. The action labels are based on the BABEL dataset, a large annotation dataset which is coupled with AMASS. Most interesting are dynamic actions, since expected muscle activations for simple dynamic actions are well documented and we present a short qualitative analysis based on such actions in Section 3.2.

## 3.2 Data Analysis, Validation, and Visualization

In Figure 5 (left) we explore the interrelation between different activities by investigating our simulated muscle activation time-series. To this end, we extract features from the temporal muscle activation sequences using tsfresh [12], a commonly used framework in time series analysis that extracts a feature vector based on time series characteristics such as mean, skewness, standard deviation *etc*. We chose distinct and descriptive groups of activities from the BMLmovi subset such as jumping, kicking, stepping and walking, the resulting features were normalized and clustered using FINCH [62] and visualized with h-NNE [63]. It can be observed, that activities do not only cluster together based on variations within the same category (e.g., different types of jumps, including jumping jacks), but also align closely across different categories, when they share similar motion patterns (e.g. sideways movements). This underlines the descriptive information contained in our simulated muscle activation sequences for characterizing activities.

## 4 Motion to Muscle Activation Estimation Benchmark

While OpenSim provides a means for simulating muscle activations, it is both highly compute intensive as well as sensitive regarding hyper parameters as described in Section 6. These properties limit it to be used by experts in an offline manner and prevent usage in everyday applications. In this section we explore the usage of MINT as a training dataset for the estimation of muscle activation based on pose motion. Such networks provide muscle activation estimation in an instant and can easily be deployed for various downstream tasks.

Given pose motion sequences, we use the preprocessing step defined by [28] which adjusts skeletal structure to a uniform format and normalizes positions and enriches the resulting data points with additional features. This procedure maps each input to a 263-dim descriptor, resulting in samples of the form $x = [x_1, ..., x_T]$, $x_t \in \mathbb{R}^{263}$. For training our models we segment the resulting data into clips of 1.4 second sampled at 20 frames per second, resulting in $T = 28$ input frames. Given a network $f_\Theta : \mathbb{R}^{T \times d} \mapsto \mathbb{R}^{T \times m}$ we predict $f(x) = y$ with $y = [y_1, ..., y_T]$, $m = 402$ being the

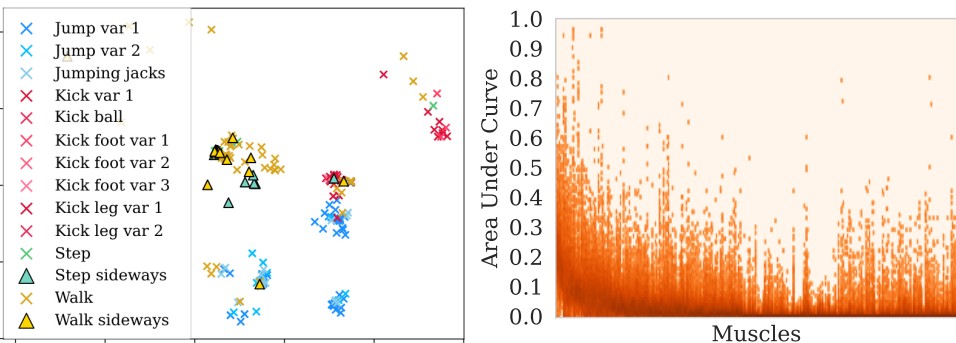

Figure 5: **Left:** Clustering of multiple activities within the BMLmovi dataset by muscle activation features. **Right:** Column-wise color coded histograms of areas under muscle activation curves for 402 muscle strains, sorted by histogram medians. Log-normalized color map, best displayed in color.

number of individual muscle strain activations simulated in our dataset, consisting of 80 lower body muscle strains from [20] and 322 muscle strains for the upper thoracolumbar region body model [3]. Evaluation is performed by calculating Root Mean Squared Error (RMSE), Pearson Correlation Coefficient (PCC), and Symmetric Mean Absolute Percentage Error (SMAPE). RMSE is commonly used but highly susceptible to data scaling, resulting in significantly lower error values for downscaled data. In practice, EMG signals vary strongly between subjects, scaling of signals is therefore a common preprocessing step. PCC is a good indicator for muscle activation series similarity, since it is scale and offset invariant. SMAPE allows for considering fixed offsets as error while being less sensitive to scaling in comparison to RMSE. PCC and SMAPE are calculated for each muscle strain individually and averaged. For our benchmark we use the train, val and test splits defined by the BABEL dataset [59]. Evaluation results are reported separately for muscles of the upper and lower body model.

## 5 Experiments

We evaluate five different architectures on MINT. Since we make use of human motion as input for our predictor, we adapted a common architecture for motion-to-motion prediction from [78] to the task of motion-to-muscle activation prediction by simply exchanging its prediction head. We further evaluate a Long Short-Term Memory (LSTM) [31], a fully convolutional network (FConv) [26], a Mamba2 Mixer model [14] and a simple transformer architecture [71] with 16 transformer layers, results for the lower and upper body model are listed in Table 2. All models are trained from scratch for 300k iterations with a batch size of 256 unless noted otherwise. More details on the model implementations can be found in the supplementary.

The evaluated transformer architecture showed the best results as compared to the adapted VQ-VAE model, LSTM, FConv and Mamba in all metrics on all evaluated motion types. The results of the experiment also show the importance of reporting PCC and SMAPE, since the differences on RMSE are marginal while PCC shows significant improvements as does SMAPE. We suspect this to be the case, since many muscles in the human body are mostly relatively inactive unless required for specific motions. For a simple analysis of this effect, we calculated the integral for each individual ground truth muscle activation sequence in all our validation set chunks and created 402 color coded histograms that are sorted by median and vertically displayed side by side on the right hand side of Figure 5 (one column in the image is a single muscle activation integral area frequency histogram). A wide range of muscles are rarely activated, resulting in the majority of activation sequences displaying integral areas significantly below 0.1 or 0.05. This property is challenging for RMSE and SMAPE, average RMSE reports a small error, since most activations are close to zero and SMAPE reports a high percentage error, since a deviation from a close to zero value is more likely to result in a high percentage deviation. For similar reasons, the upper body model displays lower RMSE and higher SMAPE, the upper body model contains a larger number of small and rarely activated muscles in contrast to the lower body model.

Table 2: Human motion-to-muscle activation prediction results for the lower- and upper body model.

| Act | VQ-VAE [78] | | | FConv [26] | | | LSTM [31] | | | Mamba2 [14] | | | Transformer [71] | | |
|---|---|---|---|---|---|---|---|---|---|---|---|---|---|---|---|
| | R↓ | S↓ | P↑ | R↓ | S↓ | P↑ | R↓ | S↓ | P↑ | R↓ | S↓ | P↑ | R↓ | S↓ | P↑ |
| *Lower body model* | | | | | | | | | | | | | | | |
| all | 0.058 | 59.7 | 0.40 | 0.052 | 66.0 | 0.49 | 0.052 | 57.8 | 0.48 | 0.051 | 55.4 | 0.49 | **0.048** | **45.1** | **0.54** |
| jump | 0.062 | 66.7 | 0.52 | 0.053 | 68.1 | 0.66 | 0.052 | 62.2 | 0.67 | **0.051** | 60.8 | 0.68 | **0.051** | **52.3** | **0.71** |
| kick | 0.069 | 69.1 | 0.38 | 0.057 | 74.9 | 0.55 | 0.058 | 66.5 | 0.55 | 0.059 | 67.6 | 0.55 | **0.053** | **54.8** | **0.62** |
| stand | 0.056 | 60.0 | 0.42 | 0.049 | 64.4 | 0.51 | 0.050 | 58.2 | 0.51 | 0.049 | 55.1 | 0.52 | **0.046** | **45.0** | **0.58** |
| walk | 0.053 | 57.7 | 0.66 | 0.046 | 61.7 | 0.73 | 0.045 | 53.7 | 0.73 | 0.045 | 50.4 | 0.74 | **0.044** | **42.4** | **0.77** |
| jog | 0.059 | 64.8 | 0.58 | 0.052 | 69.1 | 0.66 | 0.050 | 61.5 | 0.68 | 0.047 | 58.2 | 0.69 | **0.046** | **51.1** | **0.71** |
| dance | 0.070 | 71.4 | 0.40 | 0.064 | 76.0 | 0.59 | 0.063 | 71.5 | 0.57 | 0.063 | 70.2 | 0.57 | **0.057** | **58.5** | **0.65** |
| *Upper body model* | | | | | | | | | | | | | | | |
| all | 0.041 | 115.3 | 0.32 | 0.034 | 114.8 | 0.47 | 0.035 | 111.1 | 0.48 | 0.034 | 112.2 | 0.50 | **0.033** | **107.7** | **0.55** |
| jump | 0.064 | 118.1 | 0.38 | **0.052** | 119.6 | 0.54 | 0.054 | 115.4 | 0.56 | 0.053 | 117.2 | 0.58 | **0.052** | **112.7** | **0.63** |
| kick | 0.058 | 122.2 | 0.35 | 0.048 | 121.5 | 0.55 | 0.048 | 118.1 | 0.57 | 0.048 | 119.4 | 0.58 | **0.044** | **114.8** | **0.65** |
| stand | 0.039 | 117.6 | 0.34 | 0.031 | 118.2 | 0.48 | 0.031 | 114.2 | 0.49 | 0.030 | 114.9 | 0.51 | **0.028** | **110.5** | **0.55** |
| walk | 0.028 | 110.2 | 0.43 | 0.021 | 109.8 | 0.55 | 0.022 | 105.6 | 0.57 | 0.020 | 106.8 | 0.59 | **0.019** | **102.6** | **0.63** |
| jog | 0.040 | 117.1 | 0.52 | 0.034 | 118.5 | 0.64 | 0.032 | 113.9 | 0.66 | 0.031 | 115.4 | 0.66 | **0.029** | **110.8** | **0.71** |
| dance | 0.046 | 127.3 | 0.29 | 0.041 | 129.5 | 0.48 | 0.044 | 126.7 | 0.48 | 0.039 | 128.2 | 0.49 | **0.036** | **121.8** | **0.59** |

R: RSME    S: SMAPE    P: PCC

To provide a more detailed analysis we list the results on the collection of all available muscle strains in the main paper, but list further evaluations on carefully chosen subsets of major motion inducing body muscles in the appendix. We recommend future users of our dataset to consider actively evaluating on either the full range of provided muscle activations or choosing one of these muscle strand subsets depending on their specific application. Please also see the appendix for additional experiments as well as a comparison to the work of [10].

## 5.1 Qualitative Results

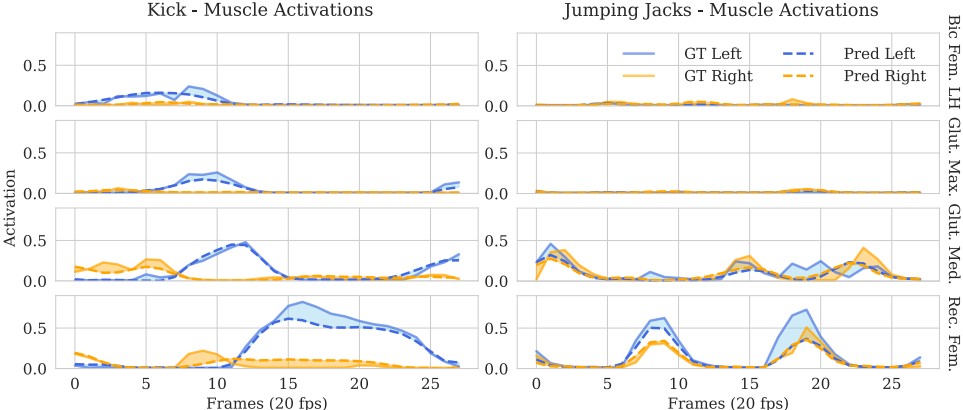

Figure 6: Example lower body muscle activations (split in left and right muscle strands) for the actions *kick* and *jumping jacks*. It is clearly visible that the kick is performed with the left leg. During *jumping jacks*, *gluteus medius* and *rectus femoris* are activated alternatingly for both legs.

In Figure 6 we list two examples from our dataset, one displaying the action *kick*, the other displaying the action *jumping jacks*, predictions are calculated with the 8-layer transformer architecture. The figure displays four key muscles essential for lower body locomotion; *biceps femoris long head* (knee flexion and hip extension), *gluteus maximus* (hip extension and external rotation), *gluteus medius* (abduction and medial rotation of the hip), and *rectus femoris* (hip flexion and knee extension), each for the left and right body half. The kick is clearly executed with the left leg with *rectus femoris* providing the force for the swing in the second half of the motion and the other muscles of the

left leg preparing it in the first half. During *jumping jacks*, *gluteus medius* and *rectus femoris* are activated alternatingly for both legs. Predicted muscle activations closely follow the ground truth from our dataset, with slight underestimation at the activation peaks. Similar estimation quality can be observed across the test set and we refer the reader to the appendix where we provide a larger number of randomly selected results for qualitative analysis.

## 6    Discussion

We believe that enhancing models through detailed muscle activation data aligned with human motion is a worthwhile direction to explore in the future, which is now made possible by the presented *MinT* dataset. The dataset offers a large amount of intricately simulated data, based on real human motions, and utilizing bio-mechanically validated musculoskeletal models. By showing that neural models can learn to connect motion input to muscle activation sequences, we broaden the pathway towards models which understand the nuanced interplay between motion and muscles.

**Societal Impact**  While the dataset has a good balance in terms of gender distribution, ethnicity is not distributed equally, and some body-weight types are less represented, impacting the dataset diversity.

**Limitations**  *MinT* is a simulation dataset, and despite careful design of our pipeline and rigorous data analysis, a synthetic-to-real domain gap remains inevitable. Researchers should be mindful of these limitations and consider their potential impact on real-world applications. Any models or analysis based on *MinT* require appropriate validation, ideally with real-world experiments.

Our simulations are a computationally intensive process. Given the potential for non-convergence in complex movement data, we imposed an iteration limit, discarding samples which do not meet a predefined error tolerance within this range. This potentially creates a category distribution shift in comparison to AMASS, since some motion categories might generally be harder to simulate.

Furthermore, the dataset is mostly restricted to motion types limited to foot-ground contact alone; motions with environment contact by other body parts or interactions with external objects were mostly excluded due to missing information about such reaction forces. We included certain object-related motions, such as lifting and throwing, as these motions are especially valuable for examining back muscle activation. Since we miss information about object mass, we assume interaction with very small, lightweight objects of negligible weight in these cases.

More extensive descriptions of these design decisions and preprocessing steps are provided in the appendix, including details on runtime distribution and error handling, to offer transparency for researchers seeking to adapt or expand upon our approach.

## 7    Conclusion

The quest to analyze human motion necessitates a critical component that has been notably absent: a comprehensive biomechanical dataset. Our contribution, the Muscles in Time (MinT) dataset, addresses this gap by providing an unprecedented collection of synthetic muscle activation data. This dataset encompasses 402 distinct simulated muscle strains, all derived from authentic human movements, thus offering a vital resource for human motion research. Our methodology entails a scalable pipeline that utilizes cutting-edge musculoskeletal models to derive muscle activations from recorded human motion sequences. The culmination of this process is the MinT dataset, which also contains 9.8 hours of time series data representing muscle activations. We demonstrate that neural networks can effectively utilize this muscle activation data to discern patterns linking motion to muscle activation. This represents a significant stride towards a deeper comprehension of human motion from a biomechanical standpoint. The MinT dataset enables the research community in exploration of human motion and muscular dynamics through a data-centric approach. Our work not only enriches the field of biomechanical studies but also paves the way for future advancements in understanding the complex interplay of muscles in human movement.

**Acknowledgements**  This work has been supported by the Carl Zeiss Foundation through the JuBot project as well as by funding from the pilot program Core-Informatics of the Helmholtz Association (HGF). The authors acknowledge support by the state of Baden-Württemberg through bwHPC. Experiments were performed on the HoreKa supercomputer funded by the Ministry of Science, Research and the Arts Baden-Württemberg and by the Federal Ministry of Education and Research.

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

# A   Appendix

## A.1   Dataset Information

The main entry point to interact with our work is our project page under https://simplexsigil.github.io/mint.

**License**   The MINT dataset is built on top of the KIT Whole-Body Human Motion Database, BMLmovi, BMLrub, the EyesJapan dataset and TotalCapture. We make use of AMASS to map from the motions of these original datasets to virtual marker positions in OpenSim.

All of these datasets allow usage of their data for non-commercial scientific research:

- The license of AMASS can be found under https://amass.is.tue.mpg.de/license.html
- The License of BMLmovi and BMLrub can be found under https://www.biomotionlab.ca/movi/
- The KIT Whole-Body Human Motion Database can be used upon citation of the original work as explained here https://download.is.tue.mpg.de/amass/licences/kit.html
- The license for the EyesJapan dataset can be found under http://mocapdata.com/Terms_of_Use.html
- The license for the Total Capture dataset can be found under https://cvssp.org/data/totalcapture/

The Muscles in Time dataset is published under a CC BY-NC 4.0 license as defined under https://creativecommons.org/licenses/by-nc/4.0/. Researchers making use of this dataset must also agree to the licenses mentioned above which can add additional restrictions depending on the individual sub-dataset.

Our data generation pipeline is licensed under Apache License Version 2.0 as defined under https://apache.org/licenses/LICENSE-2.0.

Code for training our muscle activation estimation networks is licensed under the MIT license as defined under https://opensource.org/license/mit.

**Author statement**   The authors of this work bear the responsibility for publishing the MinT dataset and related code and data.

**Data structure**   The structure of the provided MinT data is intentionally kept simple. All data is saved in CSV files or pandas DataFrames stored in pickle files. In Listing 1 we display how data for an individual sample can be loaded with minimal dependencies (*joblib* and *pandas*). We provide muscle activations in a range of $[0, 1]$, ground reaction forces and effective muscle forces. Data is provided with 50 fps, each dataframe is indexed by fractional timestamps. Columns are named meaningfully, the first 80 muscles belong to the lower body model, the following 322 muscles belong to the upper body model. The first and last 0.14 seconds are cut off since the muscle activation analysis is unstable towards the beginning and end of data. Since the data is generated in chunks of 1.4 seconds and muscle activation analysis can fail to succeed due to various factors, the provided data may contain gaps identified by missing data for certain time ranges.

**The *musint* package**   To further facilitate the usage of the MinT dataset, we provide the *musint* package, a Python package that allows data to be loaded into a predefined torch dataset and allows simplified cross-referencing with BABEL dataset labels. Additionally, it includes functionality for sampling a sub-window of the data at a given framerate as well as identifying and handling any gaps in the data. A short example on the musint package usage is displayed in Listing 2.

The *musint* package can be installed via `pip install musint`. Additional insight can be found on the musint github page where we also provide a Jupyter notebook for displaying the data as well as additional information on muscle subsets:
https://github.com/simplexsigil/MusclesInTime

```
1    >>> # First download and extract the dataset.
2    >>> # Example for sample
3    >>> #'BMLmovi/BMLmovi/Subject_11_F_MoSh/Subject_11_F_10_poses'
4    >>> import joblib
5    >>> joblib.load("muscle_activations.pkl")
6           LU_addbrev_l    ...    TL_TR4_r    TL_TR5_r
7    0.14           0.016    ...       0.003       0.061
8    0.16           0.028    ...       0.005       0.070
9    0.18           0.033    ...       0.002       0.080
10   ...              ...    ...         ...         ...
11   3.74           0.024    ...       0.020       0.028
12   3.76           0.016    ...       0.009       0.004
13   3.78           0.011    ...       0.003       0.000
14
15   [183 rows x 402 columns]
16
17   >>> joblib.load("grf.pkl")
18          ground_force_right_vx    ...    ground_torque_left_z
19   0.14                  15.962    ...                     0.0
20   0.16                  10.596    ...                     0.0
21   0.18                   3.422    ...                     0.0
22   ...                      ...    ...                     ...
23   3.72                  20.337    ...                     0.0
24   3.74                  21.572    ...                     0.0
25   3.76                  22.546    ...                     0.0
26
27   [182 rows x 18 columns]
28
29   >>> joblib.load("muscle_forces.pkl")
30          LU_addbrev_l    ...    TL_TR4_r    TL_TR5_r
31   0.14           8.430    ...       0.153      11.652
32   0.16          15.345    ...       0.283      13.240
33   0.18          19.127    ...       0.143      15.240
34   ...              ...    ...         ...         ...
35   3.72          14.437    ...       1.320       3.661
36   3.74          13.993    ...       1.270       5.330
37   3.76           9.346    ...       0.577       0.847
38
39   [182 rows x 402 columns]
```

Listing 1: Simplified loading of MinT samples with joblib and pandas.

## A.2   Additional Statistics and Information

In Figure 9 we provide additional information on the data analyzed provided with Muscles in Time. Total Capture makes up a small part of the dataset with exceptionally long sequences. The Eyes Japan Dataset provides the largest contribution with 3.2h of analyzed recordings.

In Tables 3 and 4, we list larger muscle groups in the lower and upper body model as well as their function for human motion. Muscle groups or larger muscles can be represented by multiple simulated muscles, e.g. since such muscles are attached to multiple muscle locations or exert forces in varying directions. The *Gluteus Medius* muscle is an example with three simulated activations on each side of the body.

```
1  >>> # First download and extract the dataset.
2  >>> import os
3  >>> from musint.datasets.mint_dataset import MintDataset
4
5  >>> md = MintDataset(os.path.expandvars("$MINT_ROOT"))
6
7  >>> md.by_path("TotalCapture/TotalCapture/s1/acting2_poses")
8  MintData(path_id='s1/acting2', babel_sid=12906, dataset='
       TotalCapture', amass_dur=61.7, num_frames=1114, fps=50.0,
       analysed_dur=22.26, analysed_percentage=0.36, data_path='
       TotalCapture/TotalCapture/s1/acting2_poses', weight=72.1,
       height=169.2, subject='s1', sequence='acting2_poses',
       gender='male', has_gap=False, dtype=object))
9
10 >>> md.by_path("TotalCapture/TotalCapture/s1/acting2_poses").
       get_muscle_activations(time_window=(0.3,1.),
       target_frame_count=int(0.7*20.))
11       LU_addbrev_l ...   TL_TR4_r   TL_TR5_r
12 0.30          0.094 ...      0.000      0.020
13 0.36          0.094 ...      0.003      0.042
14 0.40          0.091 ...      0.000      0.027
15  ...            ... ...        ...        ...
16 0.90          0.093 ...      0.000      0.008
17 0.94          0.093 ...      0.000      0.000
18 1.00          0.094 ...      0.001      0.009
19
20 [14 rows x 402 columns]
```

Listing 2: Loading the MinT dataset with the python musint package.

### A.3   Design Choices and More Detailed Limitations

The muscle-driven simulation, based on the approach by Falisse *et al*. [20], aims to ensure that muscle and skeletal dynamics align closely with given kinematic data while minimizing muscle effort. This process involves finding a solution within the problem space that satisfies an error tolerance and the number of collocation points, which depend on the dynamics of the kinematic data. Collocation points are used to discretize the continuous kinematic and dynamic equations into a finite set of points, making the optimization problem computationally feasible. To mitigate the risk of non-convergent or non-meaningful solutions, we implemented safeguards by restricting the deviation between the kinematic information before and after the optimization problem converges.

Given the computational complexity, we decided to use 50 collocation points per second and an error tolerance of $10^{-3}$. On an Intel Xeon Gold 6230 with 96 GB RAM, processing 6 subsequences of 1.68 seconds (including 0.14 second buffers at start and end) in parallel took approximately a median time of 45 minutes. Figure 10 displays a distribution of sample-wise runtimes in a violin plot. Non-converging samples tend to have higher runtimes and can be found on the long tail on the right. To manage the impact of unsuccessful simulations on the overall runtime, we limited the optimization problem to 2500 iterations and discard a sample if the optimization does not fall within error tolerance after this time. The AMASS sequences were divided into 1.4-second segments to mitigate a nonlinearly increasing runtime associated with longer motion sequences. After simulation, these segments were recombined into the original sequences, with muscle values smoothed at the connection points to ensure seamless transitions.

A challenge arose from minor variable distances between the AMASS body model and the ground, since the contact spheres provided by the OpenCap simulation are susceptible to changes in foot-ground distance. To provide similar foot-ground distances over all AMASS subjects, our pipeline automatically offsets the AMASS model depending on the lowest body marker over the time of the sequence.

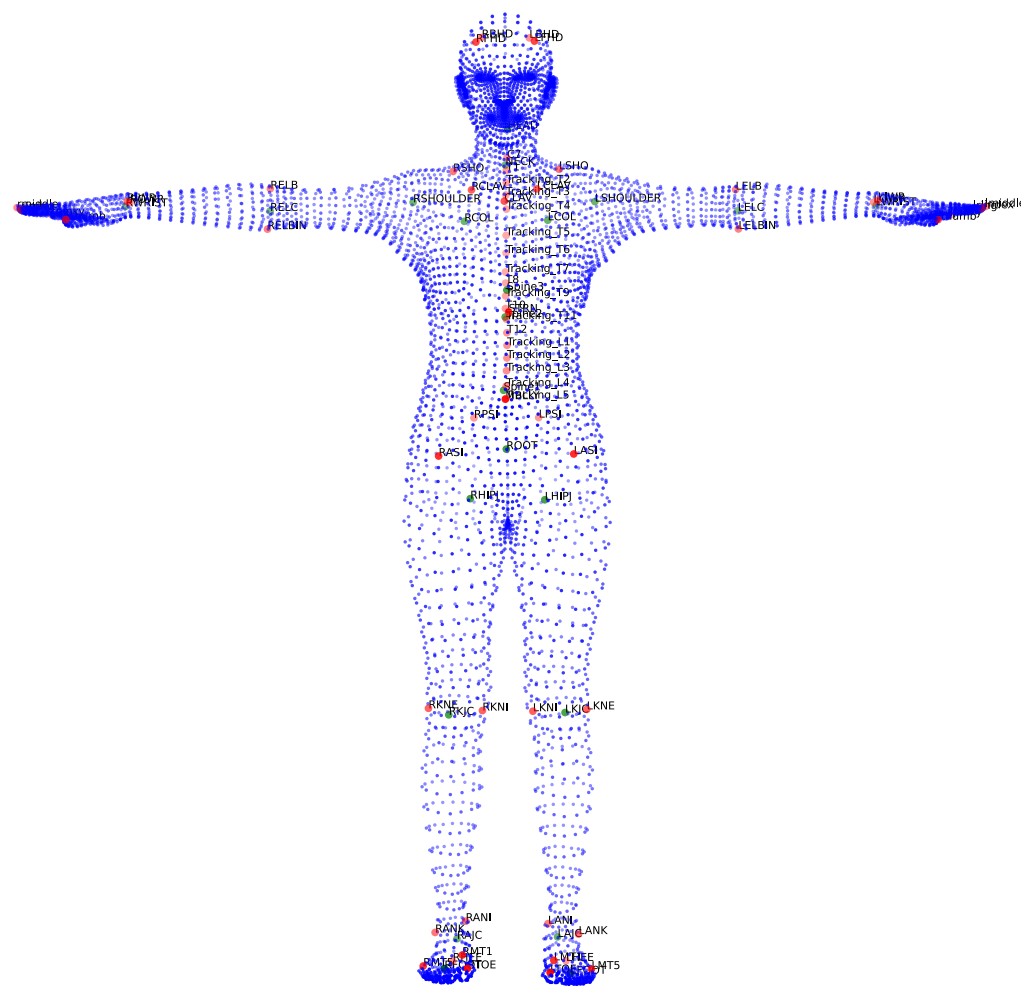

Figure 7: Virtual marker placement for transferring motions to OpenSim, enlarged from Figure 2.

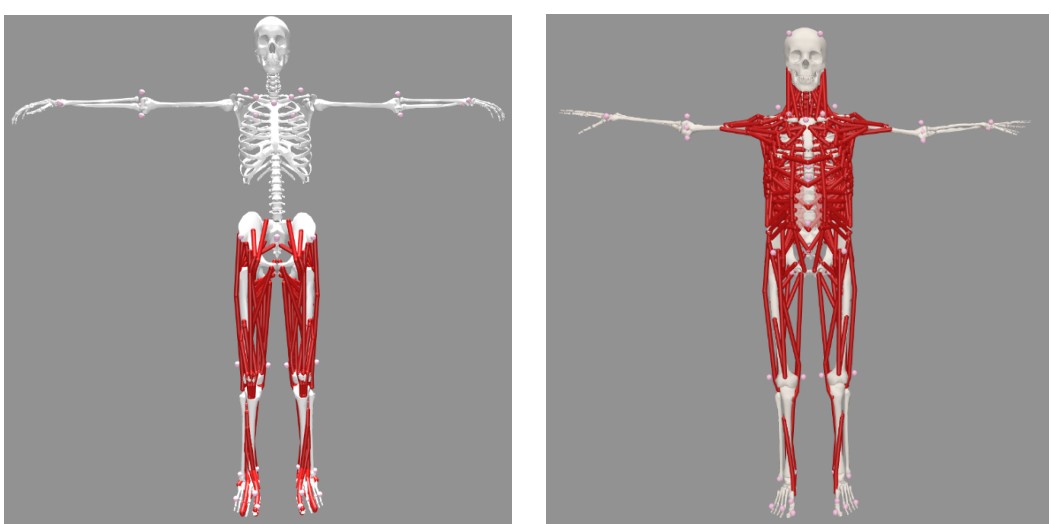

Figure 8: Lower body and upper body model, enlarged from Figure 2.

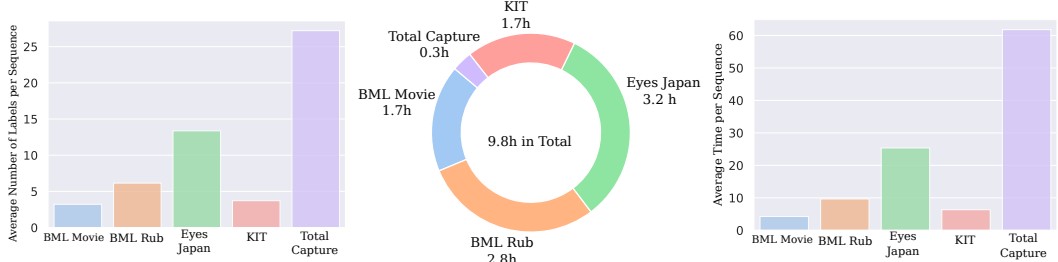

Figure 9: Average number of labels per sequence, composition of sub datasets and average sequence length.

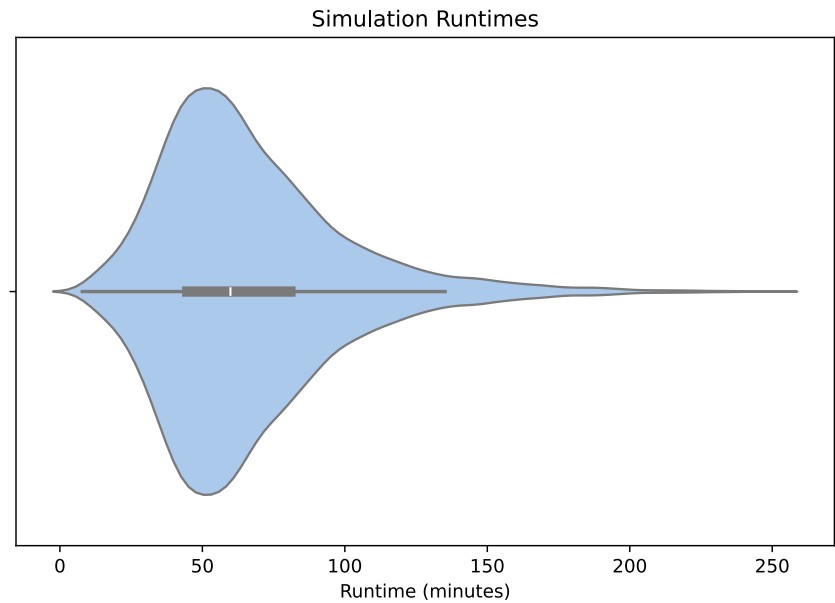

Figure 10: Analysis runtime distribution of the optimal trajectory problem described by Falisse *et al.* [20]. Subset of 10k runs.

Mapping AMASS motions to OpenSim models presented difficulties due to the numerous degrees of freedom in the Thoracolumbar model, complicating kinematic analysis. To safeguard the vertebral joints against aberrant movements, we constrained the range of motion for each vertebra, approximating the natural degrees of freedom in the vertebrae joints.

The MinT dataset was restricted to motions involving foot-ground contact only. Motions involving ground contact of other body parts or involving objects were excluded, except for motions that included throwing and lifting, which are particularly relevant for analyzing back muscle activation. In these cases, we assumed the objects' mass to be negligible, as the AMASS dataset does not provide this information.

## A.4 Results for Additional Muscle Subsets

To facilitate comparability to real world recordings as well as to other datasets, we define two muscle subsets of the lower body model, containing either 16 or eight of the most important lower body muscles for human locomotion. The subset `LAI_ARNOLD_LOWER_BODY_16` contains *left gluteus medius 1*, *left adductor magnus ischial part*, *left gluteus maximus 2*, *left iliacus*, *left rectus femoris*, *left biceps femoris long head*, *left gastrocnemius medial head*, *left tibialis anterior*, *right gluteus medius 1*, *right adductor magnus ischial part*, *right gluteus maximus 2*, *right iliacus*, *right rectus*

Table 3: List of muscle groups modelled in the model by Lai et al. [40], which are analysed in the presented approach, and their functions [82].

| Muscle | Function |
| --- | --- |
| Gluteus Maximus | Extension and rotation of the hip. |
| Gluteus Medius | Abduction and rotation of the thigh. |
| Gluteus Minimus | Abduction and rotation of the thigh. |
| Adductor Brevis | Adduction, flexion, and rotation of the thigh. |
| Adductor Longus | Adduction and flexion of the thigh. |
| Adductor Magnus | Adduction, flexion and rotation of the thigh. |
| Gracilis | Adduction, flexion and rotation of the thigh. |
| Semitendinosus | Flexion and rotation of the knee, as well as extension of the hip. |
| Semimembranosus | Flexion and rotation of the knee, as well as extension of the hip. |
| Tensor Fasciae Latae | Abduction and rotation of the thigh, as well stabilisation of the pelvis. |
| Piriformis | Rotation and extension of the thigh and abduction of thigh. |
| Sartorius | Flexion, abduction, and rotation of the hip and flexion of the knee. |
| Iliacus | Flexion of the hip. |
| Psoas | Flexion and rotation of the hip. |
| Rectus Femoris | Flexion of hip and extension of knee. |
| Biceps Femoris | Flexion of knee and extension of hip. |
| Medial Gastrocnemius | Flexion of foot and flexion of knee. |
| Lateral Gastrocnemius | Plantar flexion and knee flexion. |
| Tibialis Anterior | Dorsiflexion and inversion of the foot. |
| Vastus | Extension of the knee. |
| Extensor Digitorum Longus | Extension of toes and dorsiflexion of the foot. |
| Extensor Hallucis Longus | Extension of the big toe and dorsiflexion of the foot. |
| Flexor Digitorum Longus | Flexion of toes, as well as plantar flexion and inversion of the foot. |
| Flexor Hallucis Longus | Flexion of toes, as well as plantar flexion and inversion of the foot. |
| Peroneus (Fibularis) | Plantar flexion and eversion of the foot. |
| Soleus | Plantar flexion of the foot. |

*femoris*, *right biceps femoris long head*, *right gastrocnemius medial head* and *right tibialis anterior* while the muscle subset LAI_ARNOLD_LOWER_BODY_8 contains *left gluteus medius 1*, *left gluteus maximus 2*, *left rectus femoris*, *left biceps femoris long head*, *right gluteus medius 1*, *right gluteus maximus 2*, *right rectus femoris* and *right biceps femoris long head*. These subsets are also defined within the musint package.

In Table 5 we list the results of our 16 layer transformer model on these subsets.

Table 4: List of muscle groups modelled in the model by Bruno et al. [3], which are analysed in the presented approach, and their functions [82].

| Muscle | Function |
|---|---|
| Longissimus | Extension and rotation of the vertebrae. |
| Iliocostalis | Extension and flexion of the neck. |
| Semispinalis | Extension and rotation of the vertebrae. |
| Splenius | Extension and rotation of the vertebrae. |
| Sternocleidomastoid | Flexion and rotation of the head. |
| Scalenus | Elevation of ribs and flexion of the neck. |
| Longus Colli | Flexion of the neck and stabilisation of the cervical spine. |
| Levator Scapulae | Elevation and adduction of the scapula. |
| Quadratus Lumborum | Flexion the vertebral column. |
| Multifidus | Stabilisation cervical vertebrae. |
| Rectus Abdominis | Flexion of the lumbar spine. |
| External Oblique | Flexion and rotation of the trunk. |
| Internal Oblique | Flexion and rotation of the trunk. |
| Transversus Abdominus | Stabilisation of the trunk. |

Table 5: Human motion-to-muscle activation prediction results for the lower body model.

| Motion | All muscles | | | Lower Body | | | Subset 16 | | | Subset 8 | | |
|---|---|---|---|---|---|---|---|---|---|---|---|---|
| | RMSE↓ | PCC↑ | SMAPE↓ | RMSE↓ | PCC↑ | SMAPE↓ | RMSE↓ | PCC↑ | SMAPE↓ | RMSE↓ | PCC↑ | SMAPE↓ |
| overall | 0.036 | 0.55 | 95.3 | 0.048 | 0.54 | 45.1 | 0.066 | 0.56 | 47.7 | 0.060 | 0.56 | 45.0 |
| jump | 0.052 | 0.64 | 100.7 | 0.051 | 0.71 | 52.3 | 0.059 | 0.71 | 55.5 | 0.056 | 0.70 | 54.2 |
| kick | 0.046 | 0.64 | 102.8 | 0.053 | 0.62 | 54.8 | 0.068 | 0.63 | 57.0 | 0.067 | 0.67 | 57.4 |
| stand | 0.033 | 0.56 | 97.5 | 0.046 | 0.58 | 45.0 | 0.062 | 0.61 | 47.5 | 0.052 | 0.59 | 43.6 |
| walk | 0.026 | 0.65 | 90.7 | 0.044 | 0.77 | 42.4 | 0.060 | 0.77 | 43.3 | 0.057 | 0.77 | 43.4 |
| jog | 0.033 | 0.71 | 99.0 | 0.046 | 0.71 | 51.1 | 0.063 | 0.75 | 51.8 | 0.062 | 0.71 | 52.7 |
| dance | 0.041 | 0.60 | 109.2 | 0.057 | 0.65 | 58.5 | 0.073 | 0.66 | 59.6 | 0.072 | 0.67 | 59.5 |

## A.5 Training on Muscles in Action

We evaluate the generalizability of MinT by fine-tuning our 16-layer transformer architecture exclusively on the first and last transformer block and comparing the results with full training from scratch on Muscles in Action [10]. The motions in MIA were obtained with VIBE [39], a 3D pose estimation method performed on 2D images. The resulting motions are very noisy in contrast to the motions in AMASS which are the result of motion capture, inducing a significant domain gap. Table 6 shows our results. We find that limiting our training to the first and last transformer block results in very similar RMSE values compared to full fine-tuning, while PCC and SMAPE clearly displays a small but significant advantage of the full fine-tuning strategy. Still, finetuning the first and last layer only affects some 8% of all trainable weights, and we see this as an indication for the transferability of the knowledge obtained by training on MinT.

## A.6 Additional Qualitative Examples for MinT

Figure 6, in the main paper, lists two qualitative examples to display the muscle activation estimation quality of our best model. Additionally, Figures 11 to 17 show 48 randomly chosen samples from the test set.

Table 6: Human motion-to-muscle activation prediction results on Muscles in Action [10].

| Motion | Full Fine-tuning | | | First and last layer | | |
|---|---|---|---|---|---|---|
| | RMSE↓ | PCC↑ | SMAPE↓ | RMSE↓ | PCC↑ | SMAPE↓ |
| Overall | 15.11 | 0.27 | 37.0 | 15.15 | 0.21 | 41.6 |
| ElbowPunch | 15.66 | 0.25 | 43.6 | 15.48 | 0.19 | 48.8 |
| FrontKick | 8.49 | 0.19 | 34.5 | 8.20 | 0.14 | 41.0 |
| FrontPunch | 8.47 | 0.38 | 29.8 | 8.22 | 0.36 | 36.3 |
| HighKick | 13.09 | 0.35 | 37.0 | 12.94 | 0.29 | 39.7 |
| HookPunch | 13.18 | 0.32 | 37.1 | 13.28 | 0.28 | 44.6 |
| JumpingJack | 13.79 | 0.27 | 28.5 | 13.42 | 0.23 | 29.5 |
| KneeKick | 12.32 | 0.25 | 37.3 | 12.26 | 0.16 | 43.0 |
| LegBack | 11.70 | 0.32 | 37.3 | 11.91 | 0.18 | 44.4 |
| LegCross | 13.89 | 0.17 | 42.7 | 13.84 | 0.11 | 48.9 |
| RonddeJambe | 15.81 | 0.20 | 39.5 | 15.50 | 0.17 | 42.6 |
| Running | 7.53 | 0.30 | 26.3 | 7.25 | 0.24 | 27.4 |
| Shuffle | 9.79 | 0.21 | 28.0 | 9.56 | 0.13 | 30.5 |
| SideLunges | 26.13 | 0.29 | 45.9 | 26.66 | 0.22 | 51.7 |
| SlowSkater | 20.15 | 0.26 | 42.1 | 20.81 | 0.19 | 47.2 |
| Squat | 22.68 | 0.26 | 44.9 | 22.76 | 0.21 | 48.2 |

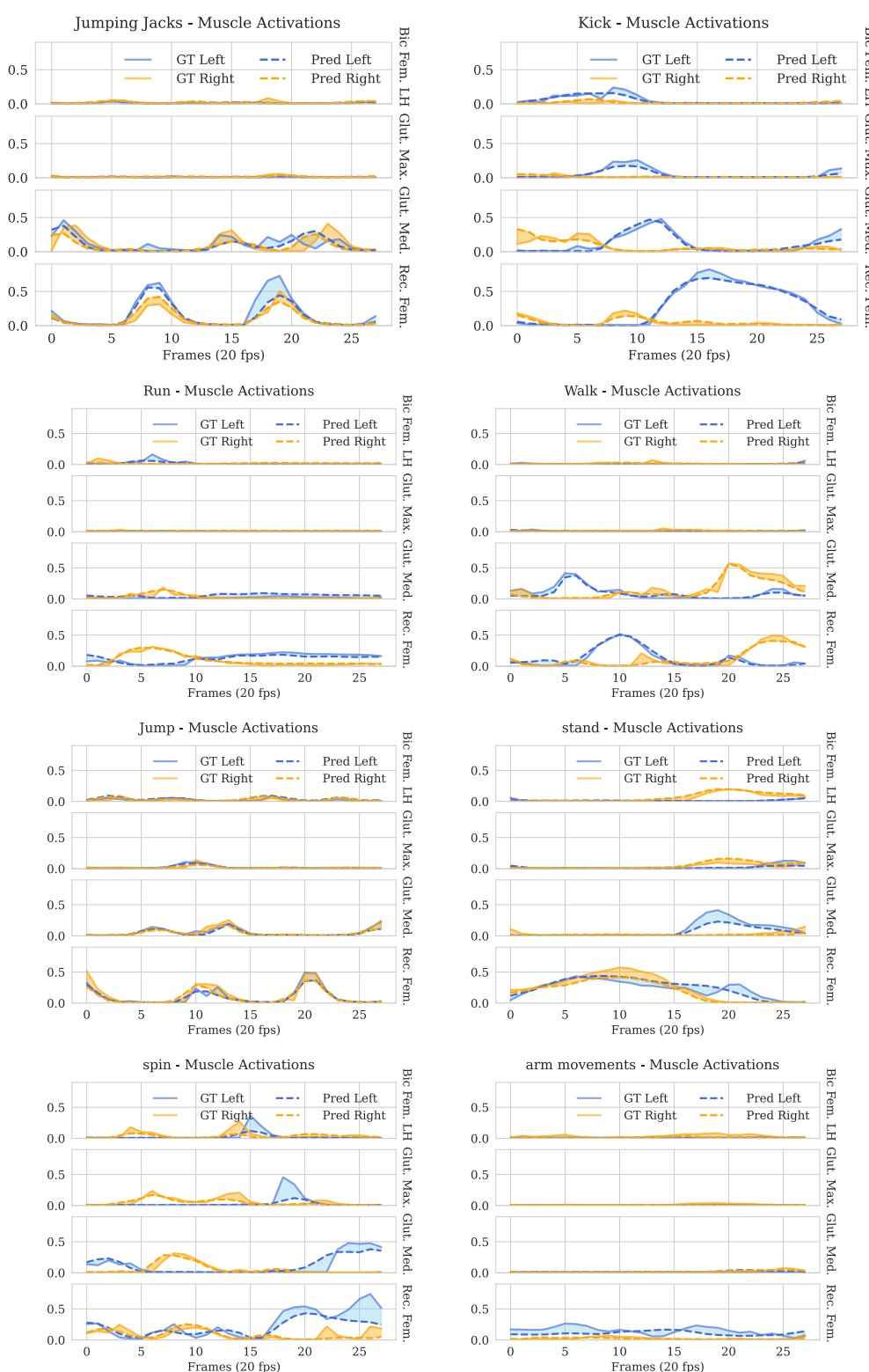

Figure 11: Muscle activation estimation with our 16 layer transformer model.

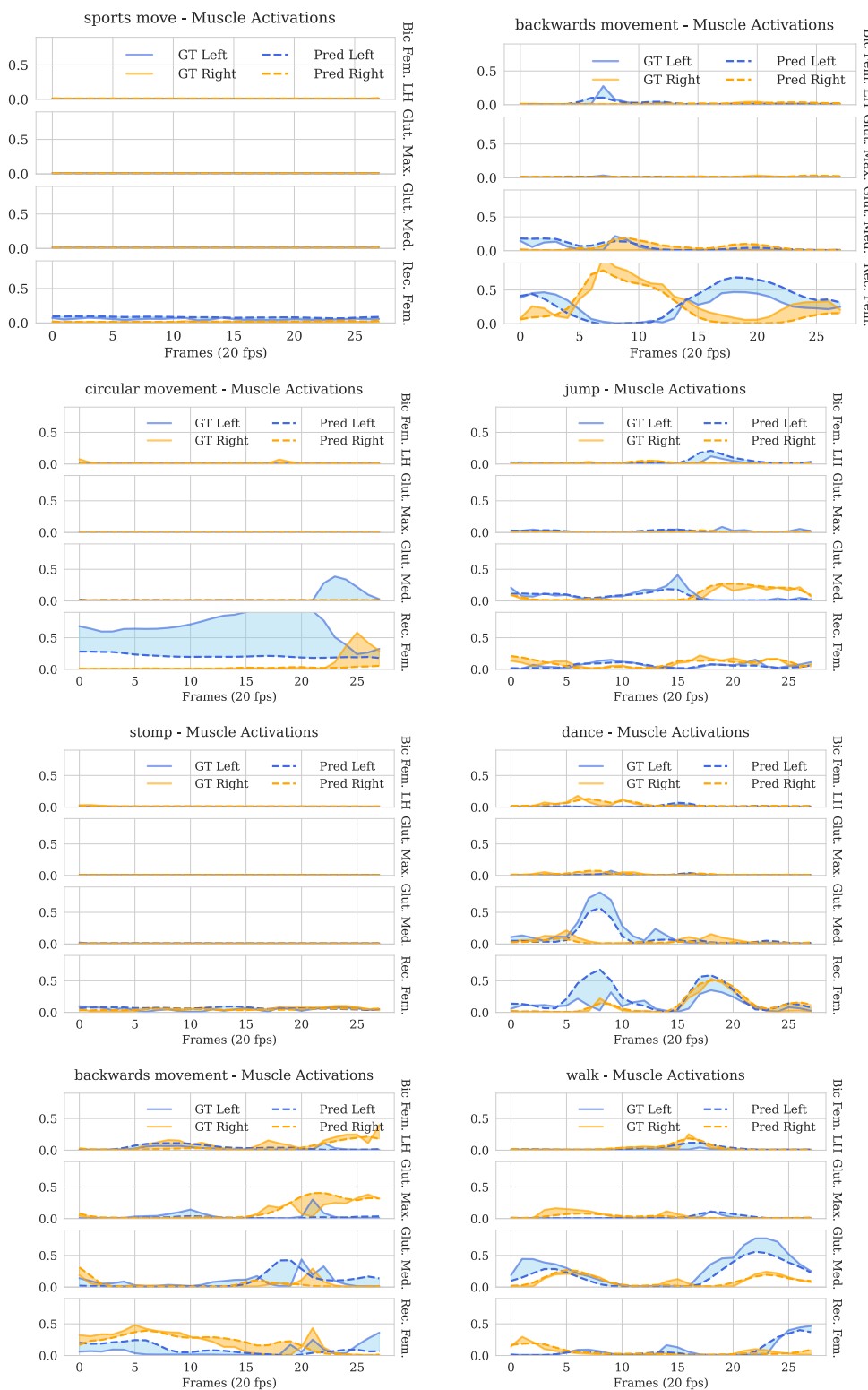

Figure 12: Muscle activation estimation with our 16 layer transformer model.

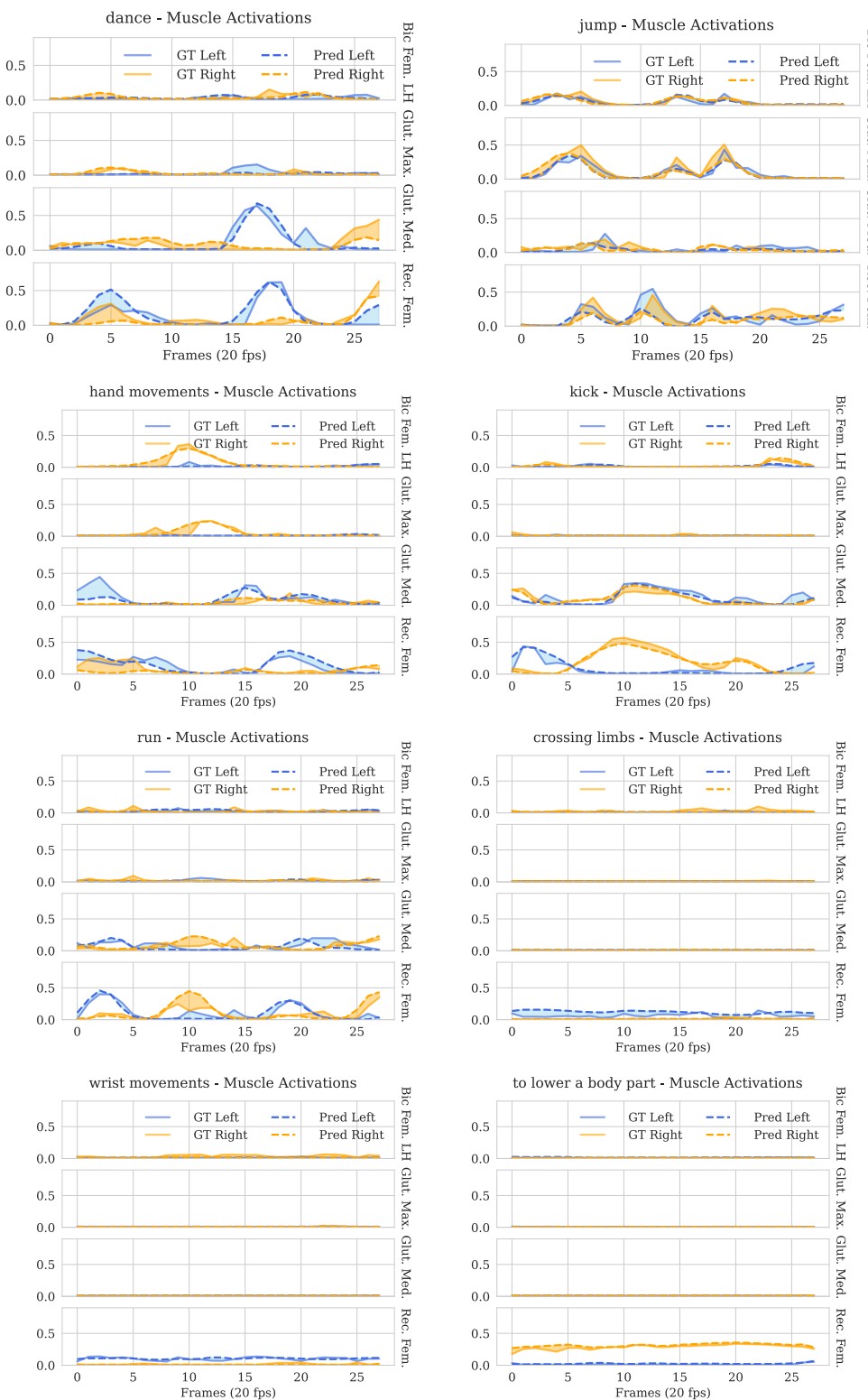

Figure 13: Muscle activation estimation with our 16 layer transformer model.

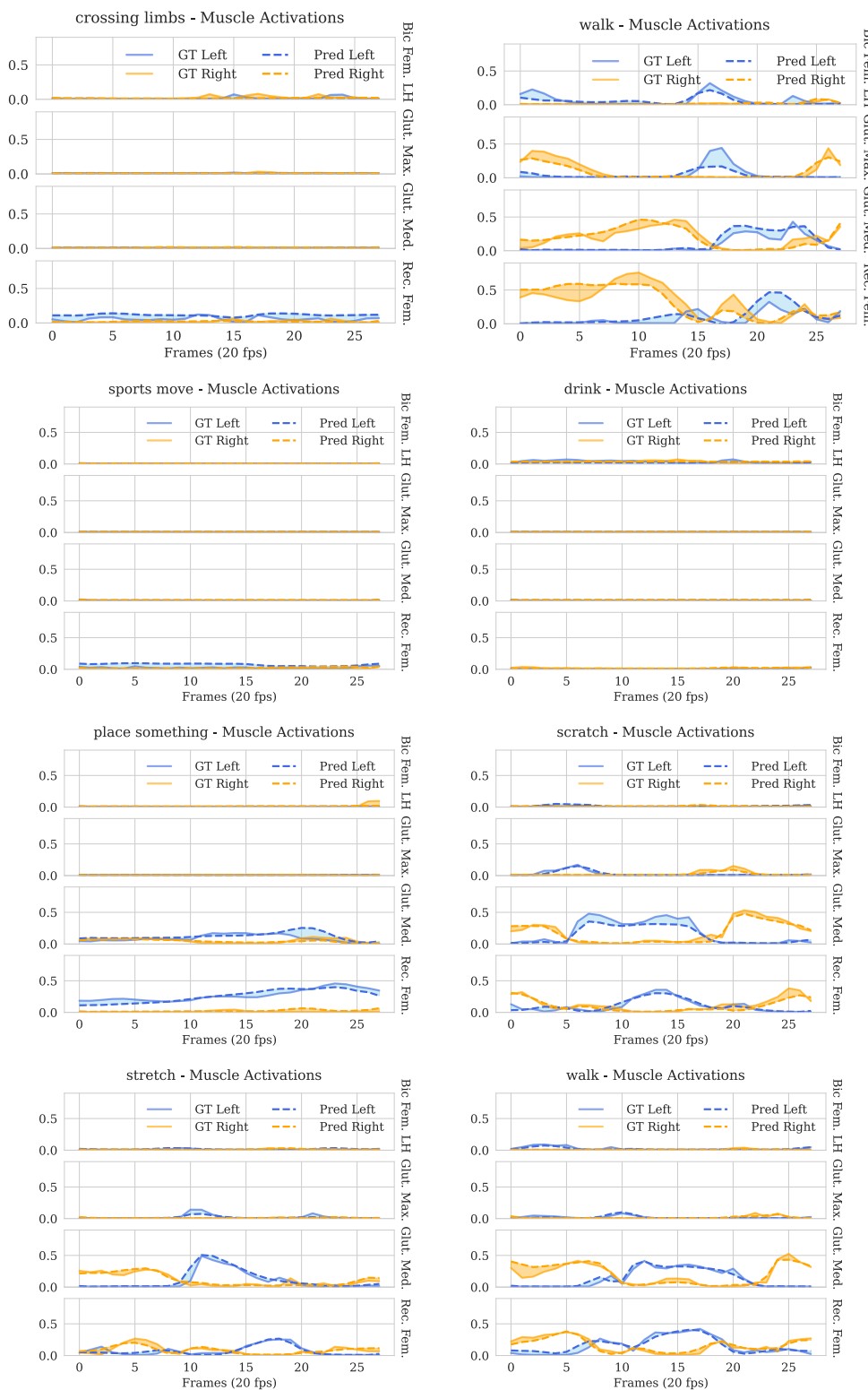

Figure 14: Muscle activation estimation with our 16 layer transformer model.

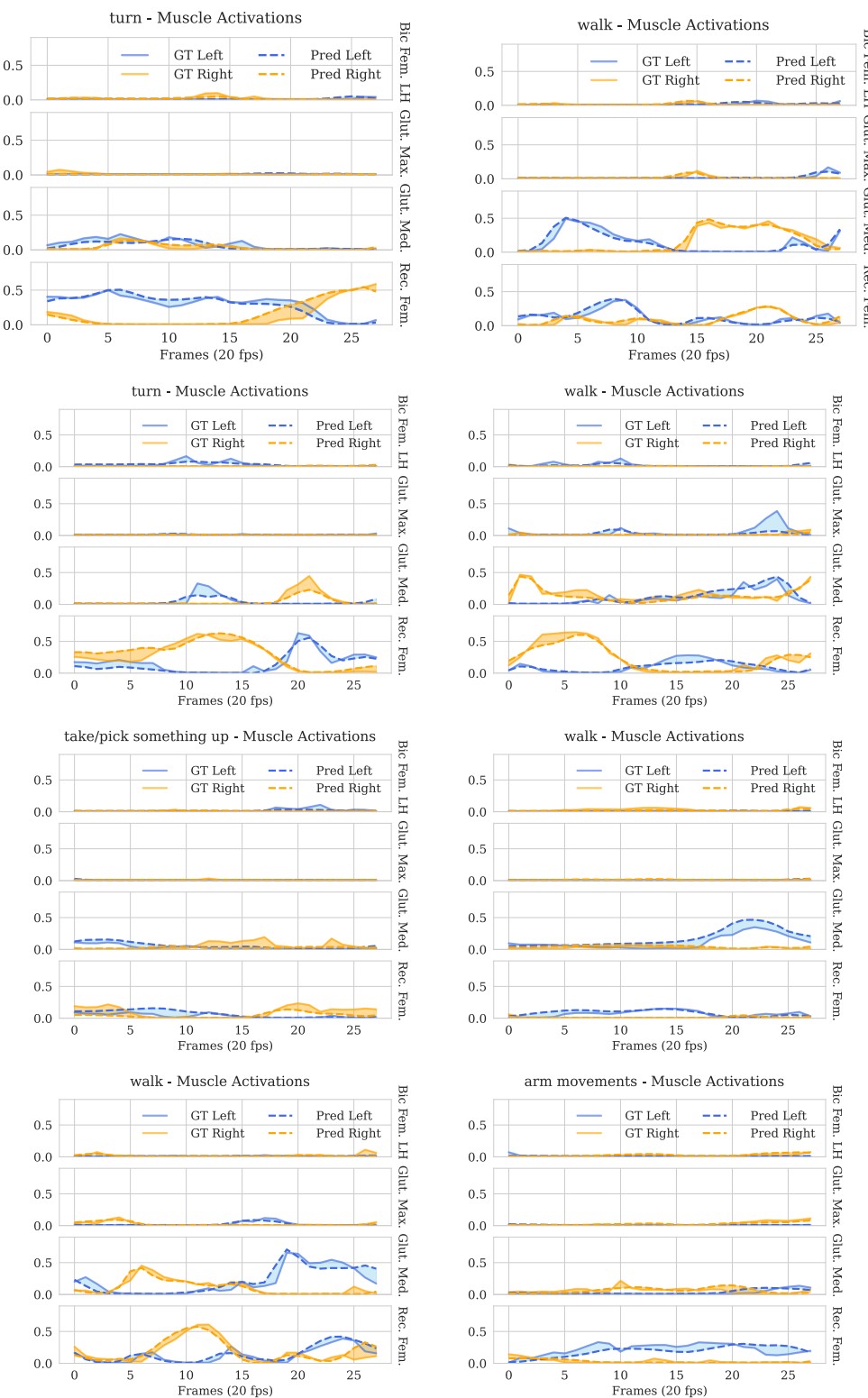

Figure 15: Muscle activation estimation with our 16 layer transformer model.

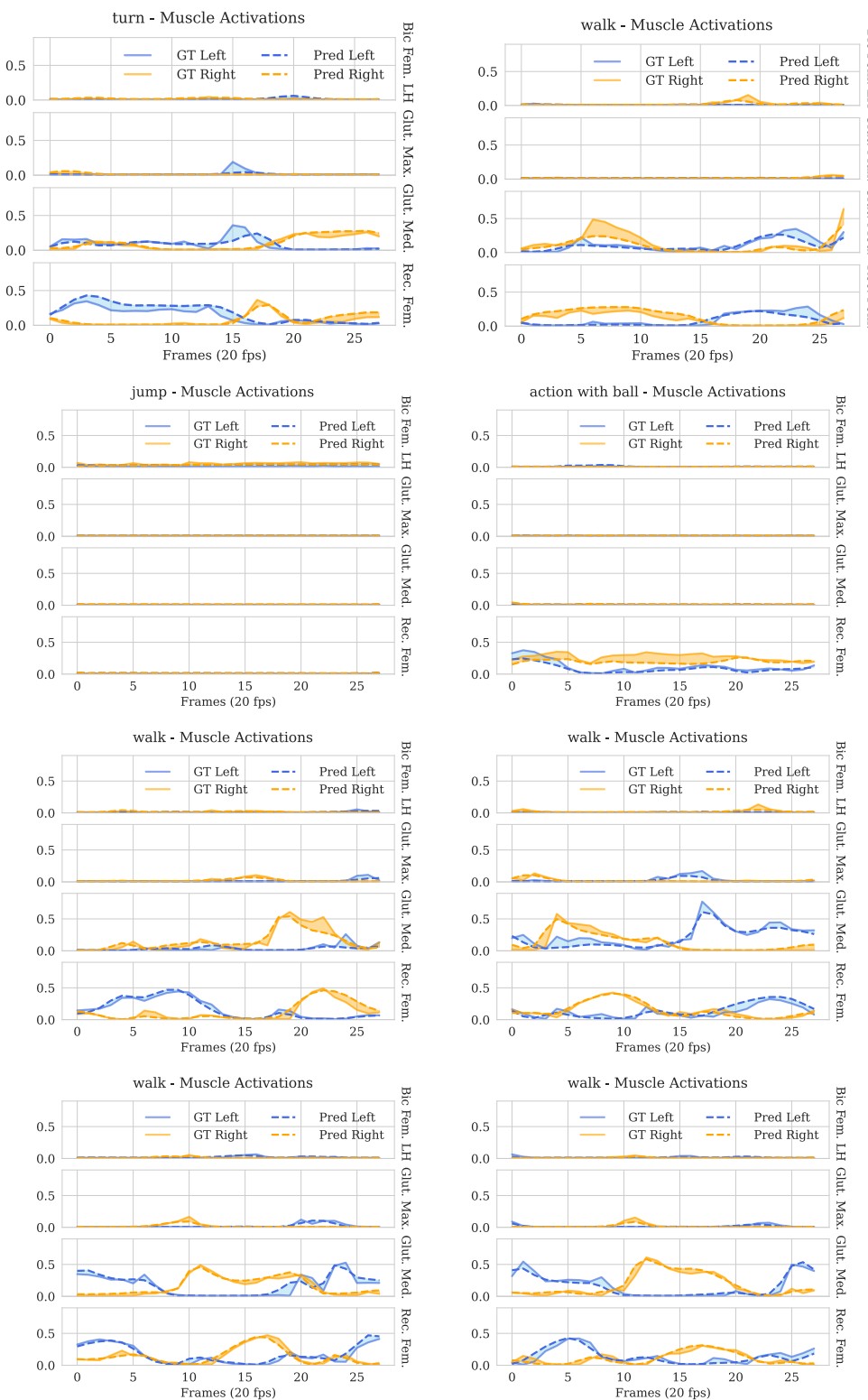

Figure 16: Muscle activation estimation with our 16 layer transformer model.

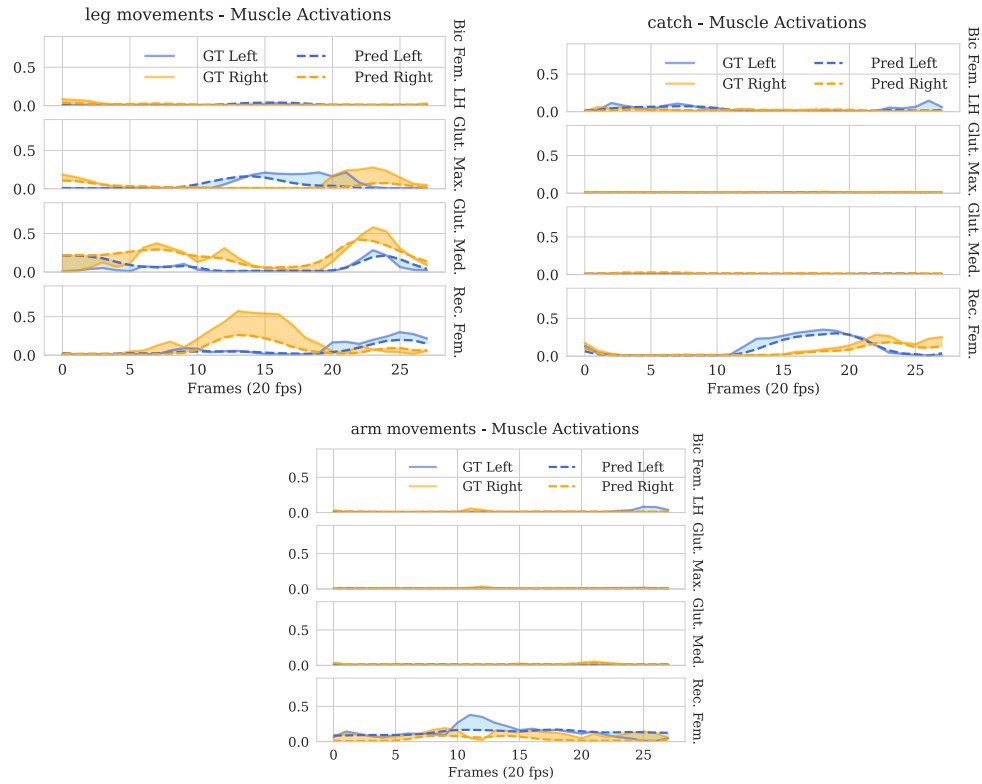

Figure 17: Muscle activation estimation with our 16 layer transformer model.

