# A  Appendix

## A.1  Dataset Information

**Dataset access and maintenance plan**  The *MinT* dataset will be provided via the persistent long-term storage service RADAR4KIT (Research Data Repository for KIT), ensuring both uninterrupted and machine readable access. Data published by *RADAR4KIT* is indexed via Metadata following the *Open Archive Initiative* interface which is automatically published to datacite.org and will automatically be referable via a DOI. Data is secured according to *Open Archival Information System* standard ISO 14721:2003 and availability is guaranteed for a minimum of 10 years.

To facilitate the review process and integrate reviewer feedback concerning the data structure (RADAR4KIT data can not be changed easily), we provide an intermediate link for direct download of our data, which will be exchanged with a RADAR4KIT link for the camera ready version.

**Currently the dataset can be downloaded under this link (2.2 GB, compressed tar file):**
https://s.kit.edu/mint-data
Our code for motion to muscle estimation can be found here:
https://github.com/simplexsigil/motion2muscle.git

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

## A.7 Corrections

In line 266 and 267 we wrote

> In the construction of the dataset, some design choices had to increase simulation robustnees, [...]

while the correct text should be

> In the construction of the dataset, some design choices were made to increase simulation robustness, [...]

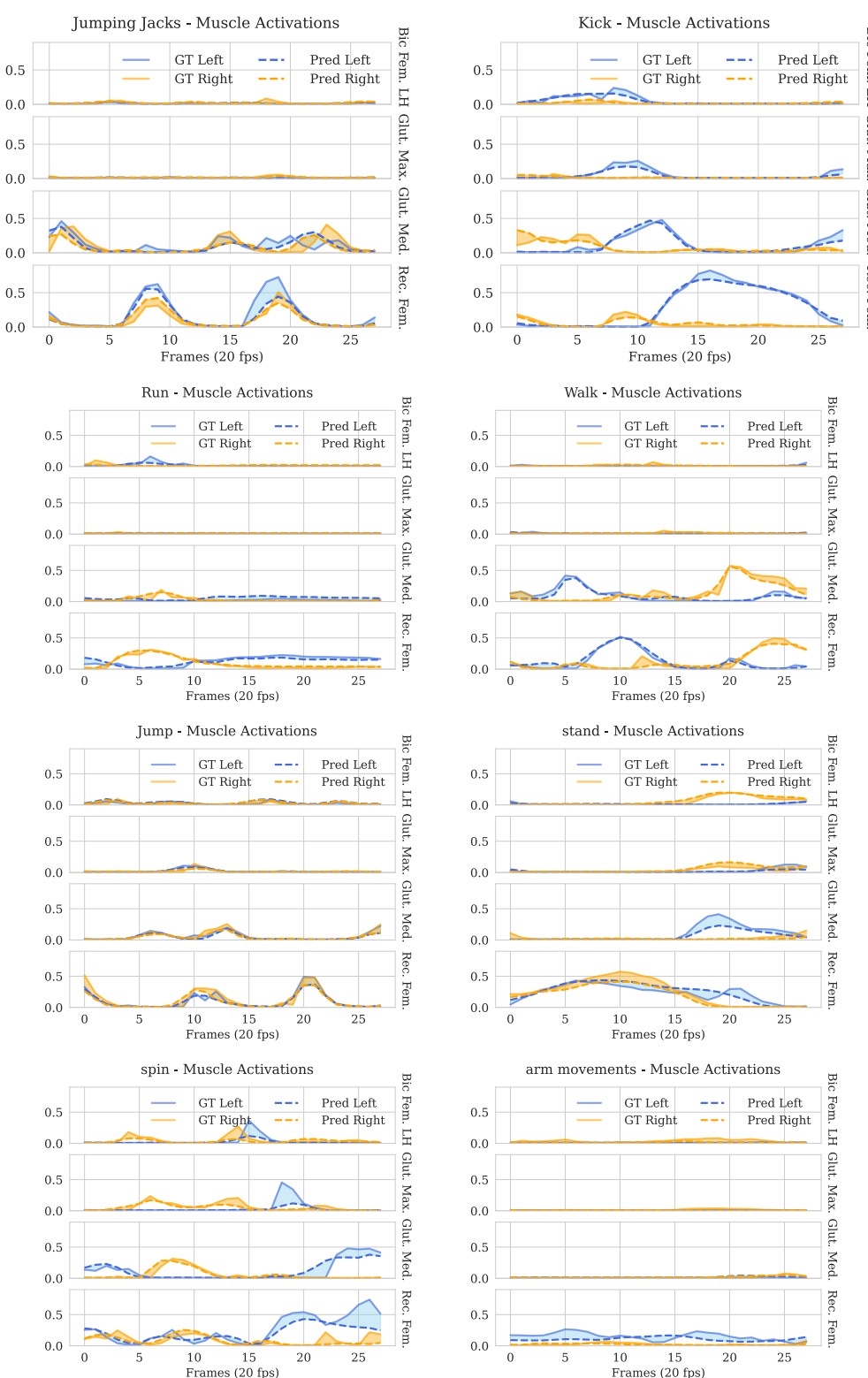

Figure 11: Muscle activation estimation with our 16 layer tranformer model.

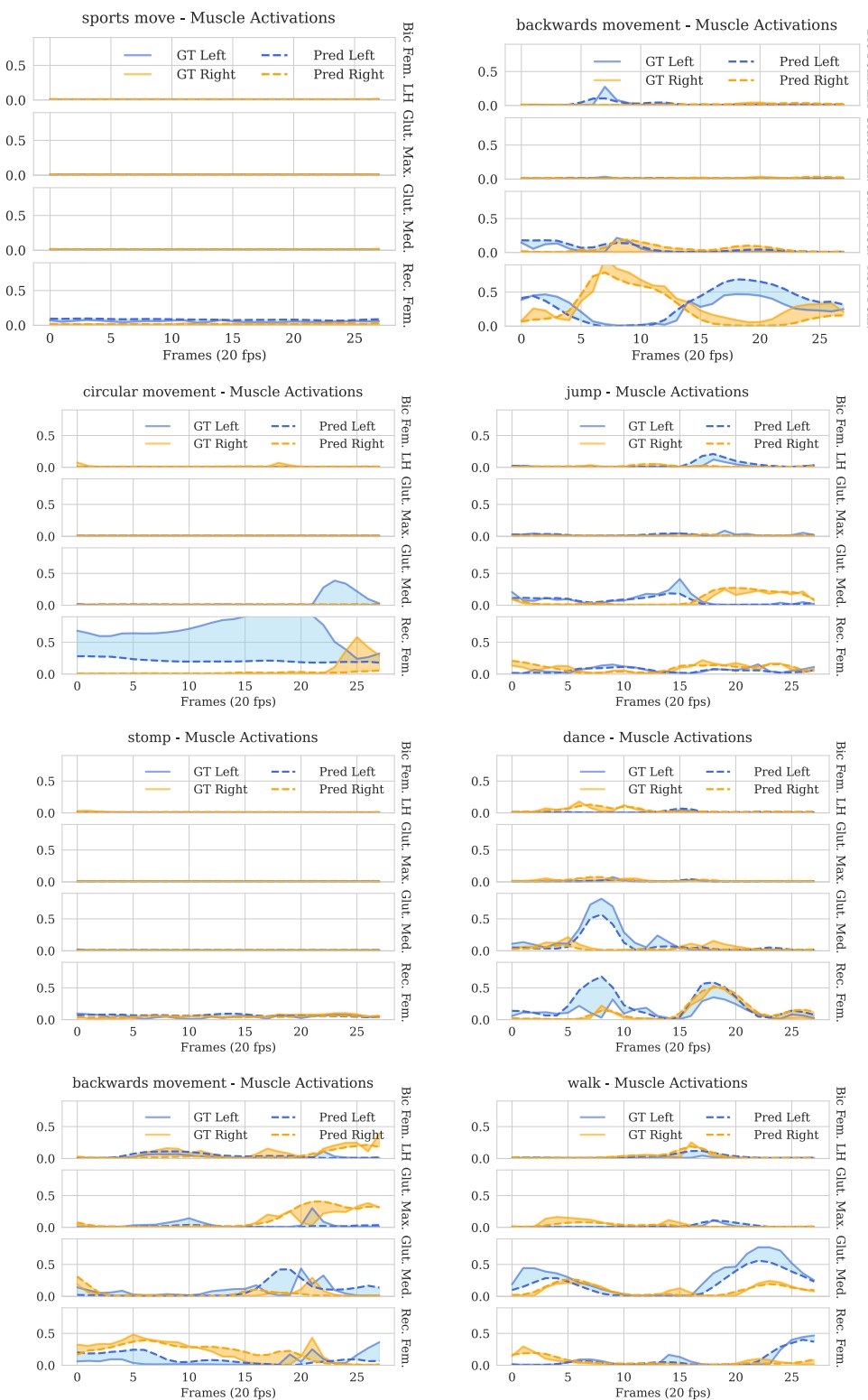

Figure 12: Muscle activation estimation with our 16 layer transformer model.

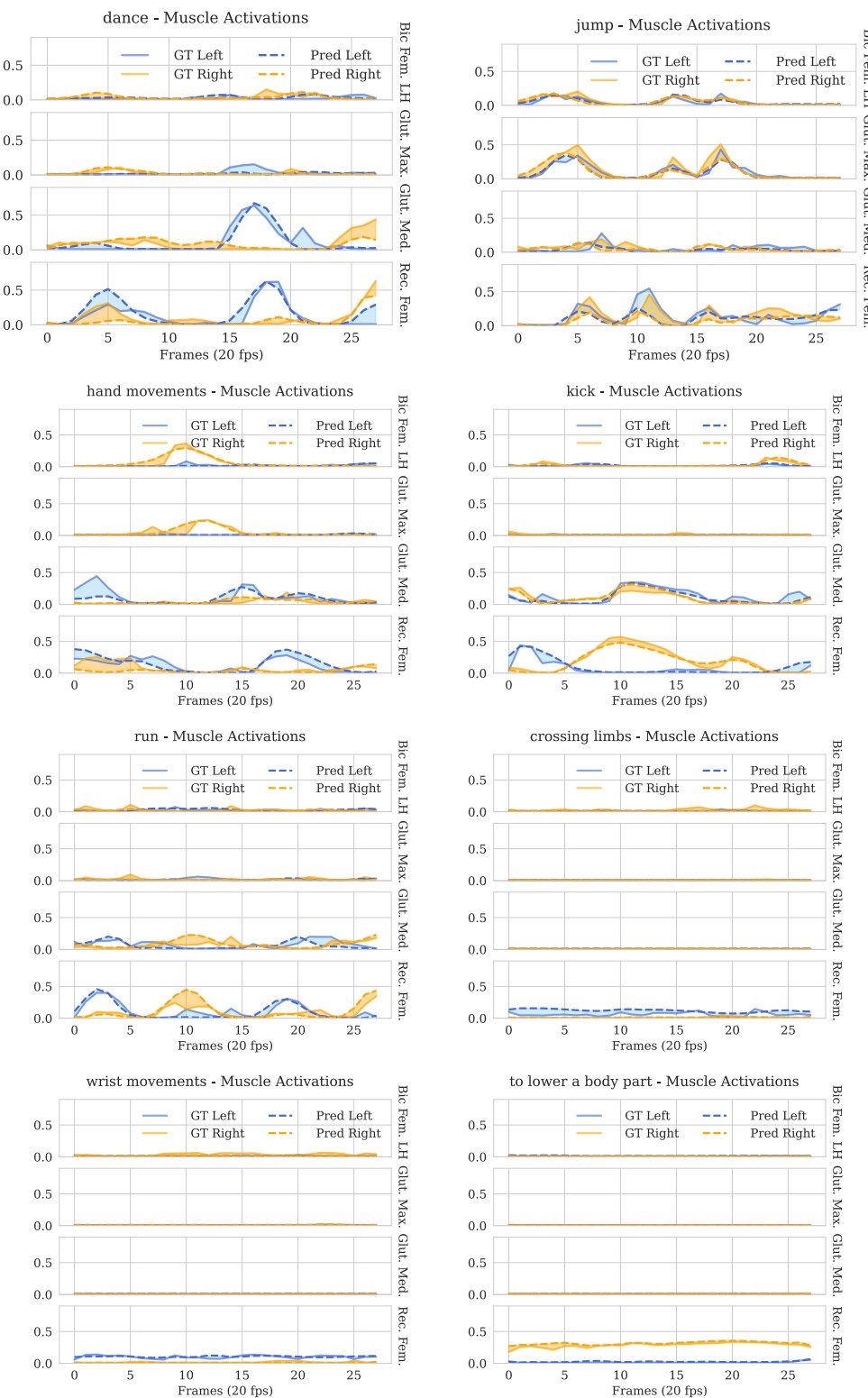

Figure 13: Muscle activation estimation with our 16 layer transformer model.

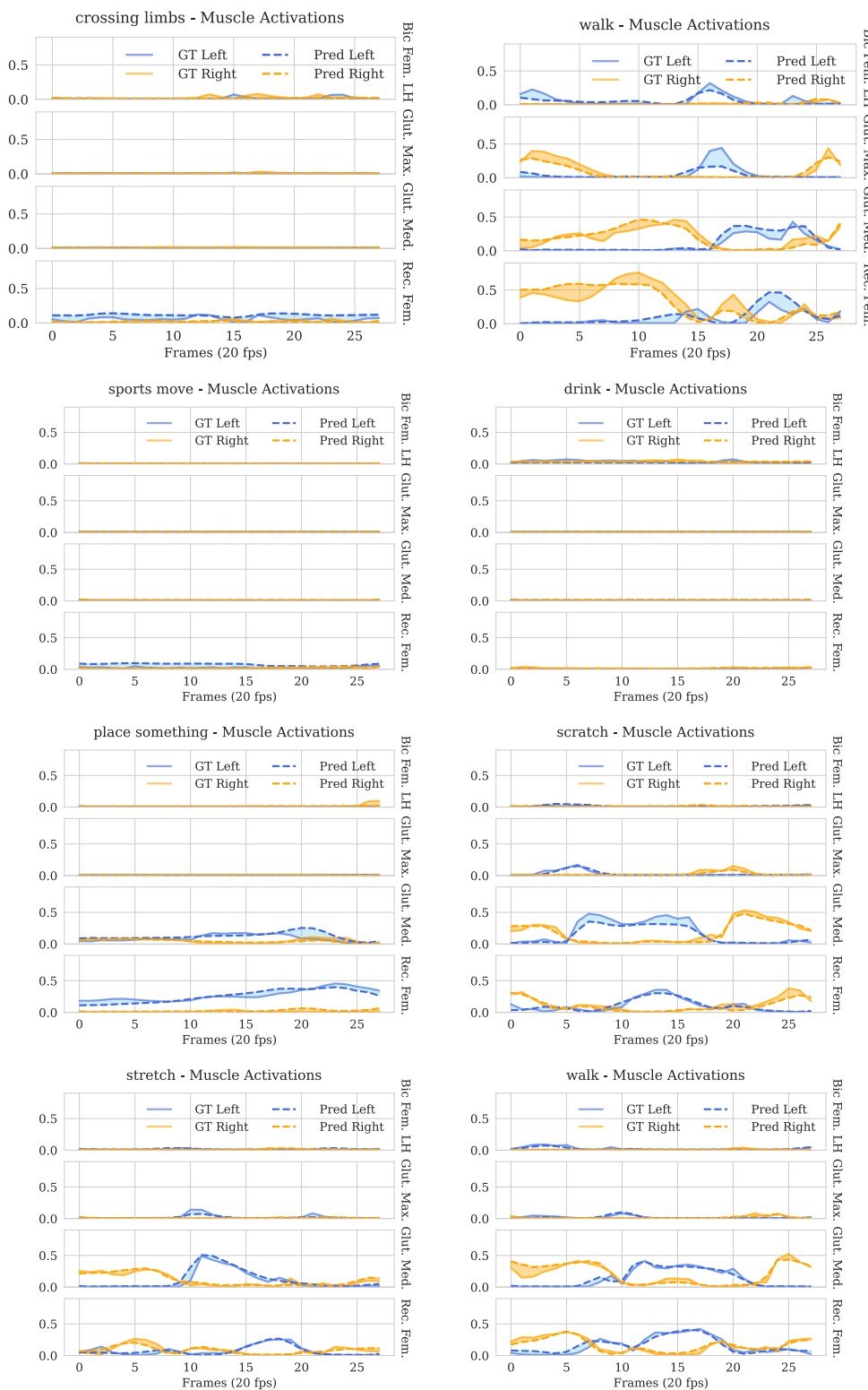

Figure 14: Muscle activation estimation with our 16 layer transformer model.

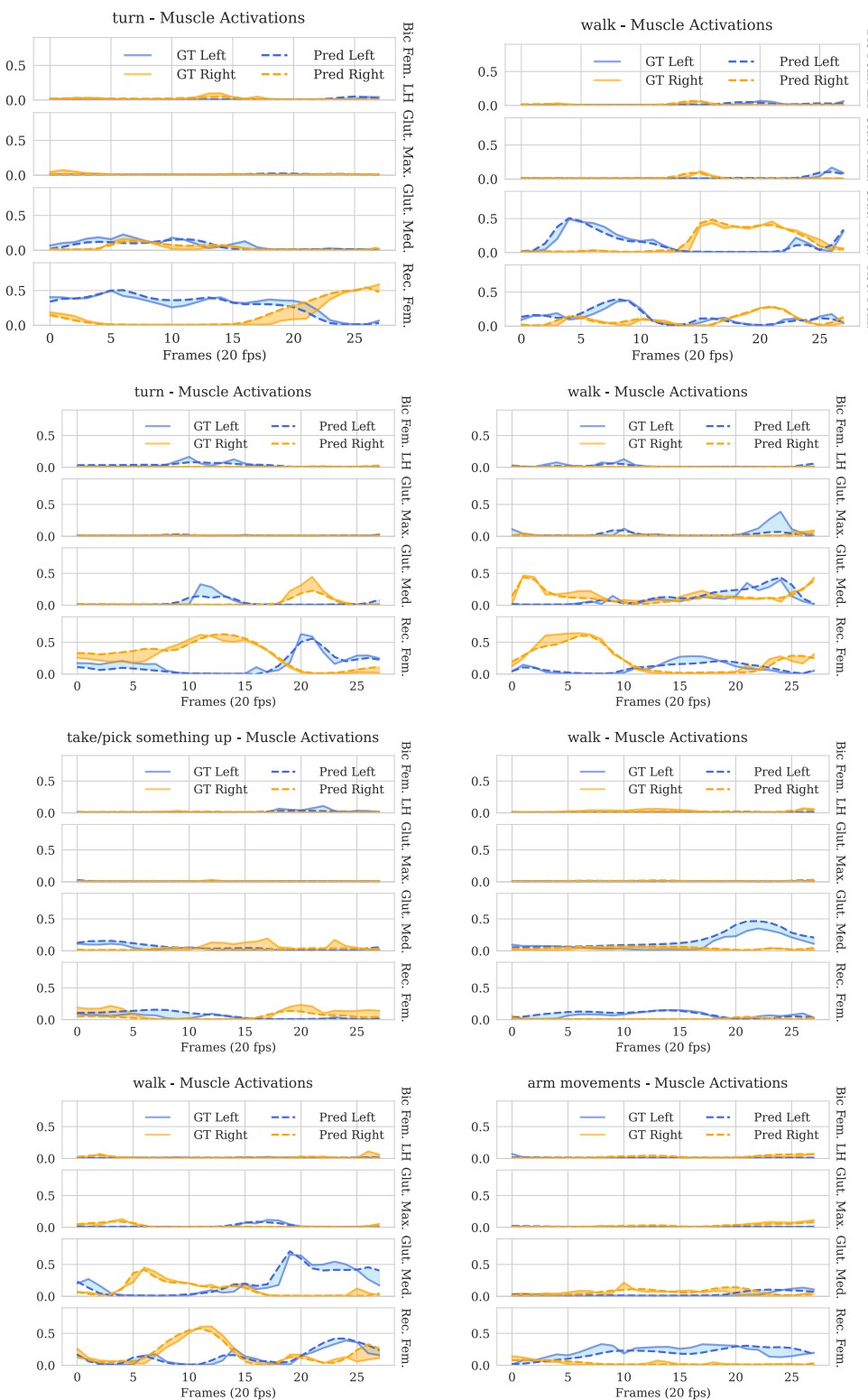

Figure 15: Muscle activation estimation with our 16 layer transformer model.

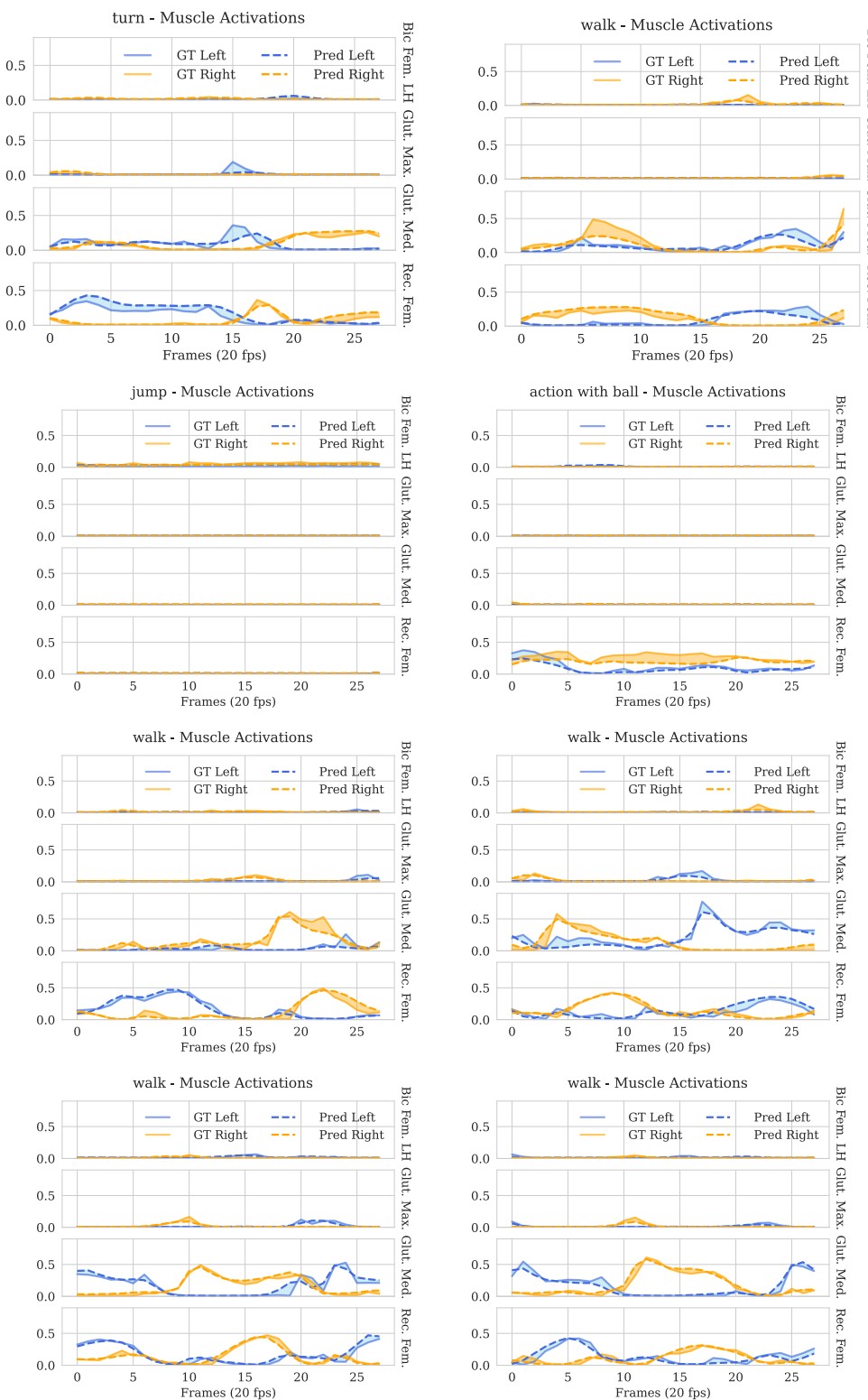

Figure 16: Muscle activation estimation with our 16 layer transformer model.

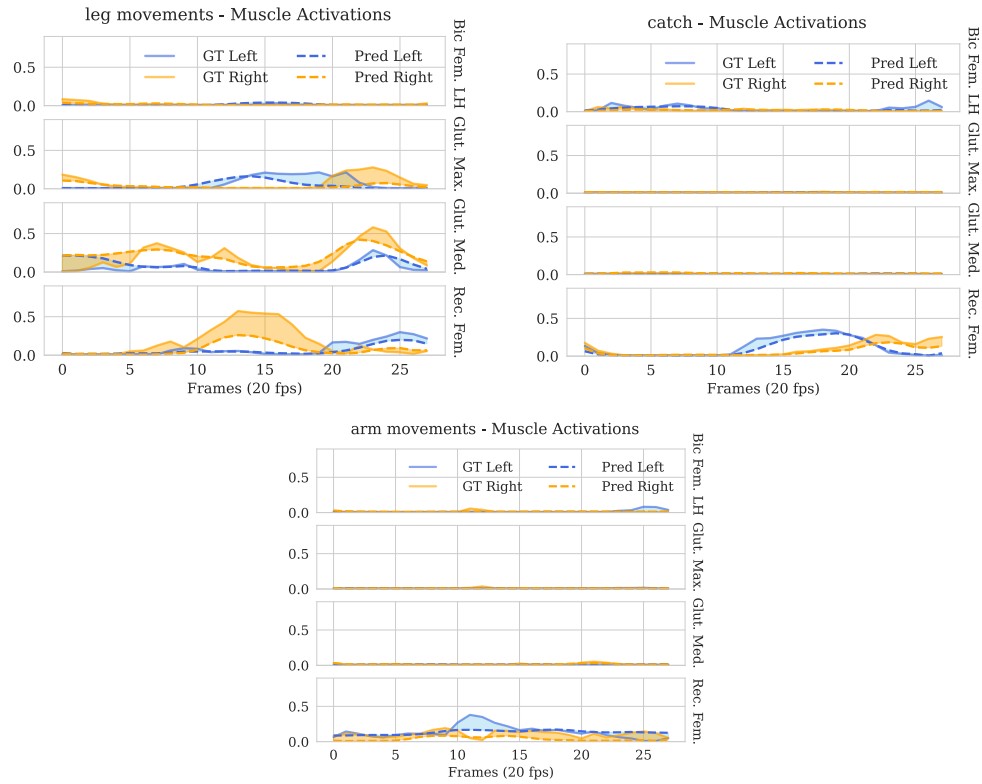

Figure 17: Muscle activation estimation with our 16 layer transformer model.