# OpenReview forum: "Muscles in Time: Learning to Understand Human Motion In-Depth by Simulating Muscle Activations"
_NeurIPS.cc/2024/Datasets_and_Benchmarks_Track — NeurIPS 2024 Track Datasets and Benchmarks Poster_

### Official Review · Reviewer_kyb6 · 2024-07-18
**Review on Submission1038**

**Rating:** 6
**Confidence:** 4
**Correctness:** The claims are correct.

**Review:**

Questions:

- Is there any quantitative evaluation of the data quality like the Hicks threshold[1]?
- The baseline selection in Sec. 5 could be insufficient. The justification of VQ-VAE from T2M-GPT as a baseline appears mainly due to the input compatibility with few insights. Why VQ-VAE is meaningful as a baseline for muscle activation learning? Rather a simple choice would be evaluating naive baselines like simple MLPs/ConvNets/VAEs, which could be light-weight counterparts for the transformer-based model. Also, Is it possible to adopt the methods in Table 1 for MinT with minor modifications?
- The experiments could be insufficient. It would help to evaluate MinT-trained models on existing datasets as listed in Table 1, providing a preliminary demonstration of the MinT generalization potential.

[1] Hicks JL, Uchida TK, Seth A, Rajagopal A, Delp SL. Is my model good enough? Best practices for verification and validation of musculoskeletal models and simulations of movement. J Biomech Eng. 2015 Feb 1;137(2):020905. doi: 10.1115/1.4029304. Epub 2015 Jan 26. PMID: 25474098; PMCID: PMC4321112.

**Strengths:**

A large-scale muscle activation dataset is meaningful for data-driven motion analysis. The efforts are appreciatable.

**Additional Feedback:**

Overall, I appreciate the efforts made to create the dataset. However, there exist some issues with the data quality, experiment sufficiency, and the limited discussion on some related works. Also, the presentation needs improvement. I would be willing to update my ratings with respect to the authors' responses.

**Clarity:**

The writings could be improved to some extent.

- For Section 3, proper subsection partitions for the data processing procedure would help.
- For Section 5, proper subsection partitions for the settings and the implementation details would help. Also, some details are missing like the loss definition.

**Documentation:**

N/A

**Limitations:**

Please refer the the Review part.

**Opportunities For Improvement:**

- For qualitative result demonstration, it would be better to supplement the corresponding motion.

**Relation To Prior Work:**

Efforts on musculoskeletal humanoid control and simulation should be also discussed.

[1] Jiang Y, Van Wouwe T, De Groote F, et al. Synthesis of biologically realistic human motion using joint torque actuation[J]. ACM Transactions On Graphics (TOG), 2019, 38(4): 1-12.

[2] Caggiano V, Wang H, Durandau G, et al. MyoSuite--A contact-rich simulation suite for musculoskeletal motor control[J]. arXiv preprint arXiv:2205.13600, 2022.

[3] Feng Y, Xu X, Liu L. MuscleVAE: Model-Based Controllers of Muscle-Actuated Characters[C]//SIGGRAPH Asia 2023 Conference Papers. 2023: 1-11.

[4] He K, Zuo C, Shao J, et al. Self Model for Embodied Intelligence: Modeling Full-Body Human Musculoskeletal System and Locomotion Control with Hierarchical Low-Dimensional Representation[J]. arXiv preprint arXiv:2312.05473, 2023.

[5] He K, Zuo C, Ma C, et al. DynSyn: Dynamical Synergistic Representation for Efficient Learning and Control in Overactuated Embodied Systems[C]//Forty-first International Conference on Machine Learning.

**Summary And Contributions:**

The authors proposed a large-scale synthetic muscle activation dataset as MinT by supplementing existing MoCap datasets with OpenSim simulated muscle activations. Neural network baselines are trained for kinematics observations-based muscle activation regression, showcasing the potential usage of MinT.

---

> ### Author Rebuttal · Authors · 2024-08-26
>
> Thank you for your thoughtful feedback and for highlighting several important areas for improvement. We appreciate the opportunity to clarify and enhance our paper based on your comments.
>
> ### **Quantitative Evaluation of Data Quality**
>
> Regarding the question if there is ***"[...] quantitative evaluation of the data quality like the Hicks threshold?"***; as also mentioned in the global rebuttal, the optimization loss used in OpenSimAD includes terms for position tracking, velocity, and acceleration errors, as well as activation minimization and reserve actuator minimization. While we did not employ specific thresholds like the Hicks threshold, we ensured that any results where the optimization did not converge within the recommended tolerance were excluded from our dataset. We recognize the importance of further quantitative evaluation and are considering how to incorporate this in future work.
>
> ### **Evaluation on Existing Datasets**
>
> You suggested to ***"[...] evaluate MinT-trained models on existing datasets as listed in Table 1, [...]"*** We attempted this with the MiA dataset, which is best suited for comparison. However, MiA presents challenges due to issues such as inaccurate pose estimation and subject-specific EMG recording noise. Consequently, fine-tuning on MiA was necessary, as detailed in Table A5 of the supplementary materials and also mentioned in the response to reviewer 6vHW and the global rebuttal. We plan to explore advanced domain adaptation methods to address these challenges in future work.
>
> ### **Qualitative Result Demonstration**
>
> Finally, in response to your suggestion for qualitative result demonstration, we have prepared videos that showcase predictions on MinT, as well as on the model from Table A5, combined with the corresponding motion. These can be accessed at [https://s.kit.edu/mint-vis](https://s.kit.edu/mint-vis) for MinT and [https://s.kit.edu/mia-vis](https://s.kit.edu/mia-vis) for MiA. The Jupyter notebooks used to generate these videos are available in the MinT repository. Based on these animations, we also provide a comparison video, available at [https://s.kit.edu/mint-mia-comparison](https://s.kit.edu/mint-mia-comparison) (video), which demonstrates the similarities in activation patterns between MiA and MinT, despite the domain gap. Such comparisons will be included in the final paper. To facilitate interpretation of the EMG patterns, we marked the lowest compression points of each leg using blue and orange dotted vertical lines for the left and right legs, respectively. Visibly, MiA exhibits significant noise and subject-specific scaling variations, while MinT demonstrates similar but more controlled and consistent patterns. This difference underscores the complexity of directly applying a model trained on MinT to MiA data, as MiA contains strong subject-specific characteristics. MiA itself does not provide a cross-subject split but rather allows a model to learn subject-specific characteristics and exposes the model to the same subjects at test time. We believe that subject specific EMG recording variations are so large that advanced domain generalization or adaptation techniques are necessary to achieve high performance with little or no training on the MiA target domain dataset when trained on other data like MinT. While this is an important area for future research, it extends beyond the scope of our current paper. We refer to the MiA fine-tuning experiment in Table A5 of the supplementary materials for additional context.
>
> ### **Clarity and Structure Improvements**
>
> You noted that ***"the writings could be improved to some extent,"*** particularly in Sections 3 and 5. We agree with your suggestion and will add subsection partitions to better organize the data processing and will add missing details regarding the implementation, including a clear definition of the loss functions.
>
> ### **Discussion of Related Work**
>
> In order to address your suggested improvements on related work, we will expand our discussion to include further publications on musculoskeletal humanoid control and simulation, such as those by Jiang et al. (2019), Caggiano et al. (2022), Feng et al. (2023) and He et al. (2023 and 2024), to better situate our work within the broader field.
>
> ### **Baseline Selection in Section 5**
>
> Regarding your concern that ***"the baseline selection in Sec. 5 could be insufficient,"*** we selected the T2M-GPT architecture because of its effectiveness in generating intermediate representations for cross-modality translation tasks in human motion, which we believe bears some similarity with our task in comparison to unrelated architectures. However, we acknowledge the value of comparing our approach to additional baselines. In response to your suggestion, we are preparing such baseline experiments and will provide them during this week.
>
> We are committed to incorporating these improvements and believe they will significantly enhance the quality of the paper. We appreciate your feedback and hope for the opportunity to address these concerns in our revision.

---

> > ### Author Rebuttal · Authors · 2024-08-28
> >
> > ### **Additional Baseline Results**
> >
> > As a further response to your concern ***"the baseline selection in Sec. 5 could be insufficient, [...] Rather a simple choice would be evaluating naive baselines [...]"***, we now provide results for the recently published Mamba2-Mixer model as well as a simple sequence-to-sequence LSTM model and a fully convolutional sequence-to-sequence model. These results relate to Table 2 in the main paper and can be compared to the VQ-VAE and transformer model listed there.
> >
> > All architectures find themselves in between the 16-layer transformer architecture and VQ-VAE on table 2 in the overall setting. Mamba2 improves over VQ-VAE, especially for the upper body, so does the LSTM model, to a lesser degree. The fully convolutional model improves over VQ-VAE on the upper body, on the lower body it results in higher SMAPE.
> >
> > We thank you for pointing out how additional baselines would benefit the paper and will add them to the paper accordingly. We are looking forward to your comments.
> >
> > ### **Mamba2 Mixer Model, 8 layers, internal dim 512, 14.37 M parameters**
> >
> > $$\\begin{array} {|r|rrr|}
> > \\hline
> > \\textbf{Motion} & \\textbf{RMSE↓} & \\textbf{PCC↑} & \\textbf{SMAPE↓} \\\\
> > \\hline
> > \\textbf{Lower Body} \\\\
> > \mathrm{Overall} & 0.051 & 0.49 & 55.4 \\\\
> > \mathrm{Jump}    & 0.051 & 0.68 & 60.8 \\\\
> > \mathrm{Kick}    & 0.059 & 0.55 & 67.6 \\\\
> > \mathrm{Stand}   & 0.049 & 0.52 & 55.1 \\\\
> > \mathrm{Walk}    & 0.045 & 0.74 & 50.4 \\\\
> > \mathrm{Jog}     & 0.047 & 0.69 & 58.2 \\\\
> > \mathrm{Dance}   & 0.063 & 0.57 & 70.2 \\\\
> > \\hline
> > \\textbf{Upper Body} \\\\
> > \mathrm{Overall} & 0.034 & 0.50 & 112.2 \\\\
> > \mathrm{Jump}    & 0.053 & 0.58 & 117.2 \\\\
> > \mathrm{Kick}    & 0.048 & 0.58 & 119.4 \\\\
> > \mathrm{Stand}   & 0.030 & 0.51 & 114.9 \\\\
> > \mathrm{Walk}    & 0.020 & 0.59 & 106.8 \\\\
> > \mathrm{Jog}     & 0.031 & 0.66 & 115.4 \\\\
> > \mathrm{Dance}   & 0.039 & 0.49 & 128.2 \\\\
> > \\hline
> > \\end{array}$$
> >
> >
> > ### **Simple LSTM Model, 8 layers, internal dim 256, 12.5 million parameters**
> >
> > $$\\begin{array} {|r|rrr|}
> > \\hline
> > \\textbf{Motion} & \\textbf{RMSE↓} & \\textbf{PCC↑} & \\textbf{SMAPE↓} \\\\
> > \\hline
> > \\textbf{Lower Body} \\\\
> > \mathrm{Overall} & 0.052 & 0.48 & 57.8 \\\\
> > \mathrm{Jump}    & 0.052 & 0.67 & 62.2 \\\\
> > \mathrm{Kick}    & 0.058 & 0.55 & 66.5 \\\\
> > \mathrm{Stand}   & 0.050 & 0.51 & 58.2 \\\\
> > \mathrm{Walk}    & 0.045 & 0.73 & 53.7 \\\\
> > \mathrm{Jog}     & 0.050 & 0.68 & 61.5 \\\\
> > \mathrm{Dance}   & 0.063 & 0.57 & 71.5 \\\\
> > \\hline
> > \\textbf{Upper Body} \\\\
> > \mathrm{Overall} & 0.035 & 0.48 & 111.1 \\\\
> > \mathrm{Jump}    & 0.054 & 0.56 & 115.4 \\\\
> > \mathrm{Kick}    & 0.048 & 0.57 & 118.1 \\\\
> > \mathrm{Stand}   & 0.031 & 0.49 & 114.2 \\\\
> > \mathrm{Walk}    & 0.022 & 0.57 & 105.6 \\\\
> > \mathrm{Jog}     & 0.032 & 0.66 & 113.9 \\\\
> > \mathrm{Dance}   & 0.044 & 0.48 & 126.7 \\\\
> > \\hline
> > \\end{array}$$
> >
> > ### **Simple fully convolutional model, 16 layers, internal dim 256, 13.2 M parameters**
> >
> > $$\\begin{array} {|r|rrr|}
> > \\hline
> > \\textbf{Motion} & \\textbf{RMSE↓} & \\textbf{PCC↑} & \\textbf{SMAPE↓} \\\\
> > \\hline
> > \\textbf{Lower Body} \\\\
> > \mathrm{Overall} & 0.052 & 0.49 & 66.0 \\\\
> > \mathrm{Jump}    & 0.053 & 0.66 & 68.1 \\\\
> > \mathrm{Kick}    & 0.057 & 0.55 & 74.9 \\\\
> > \mathrm{Stand}   & 0.049 & 0.51 & 64.4 \\\\
> > \mathrm{Walk}    & 0.046 & 0.73 & 61.7 \\\\
> > \mathrm{Jog}     & 0.052 & 0.66 & 69.1 \\\\
> > \mathrm{Dance}   & 0.064 & 0.59 & 76.0 \\\\
> > \\hline
> > \\textbf{Upper Body} \\\\
> > \mathrm{Overall} & 0.034 & 0.47 & 114.8 \\\\
> > \mathrm{Jump}    & 0.052 & 0.54 & 119.6 \\\\
> > \mathrm{Kick}    & 0.048 & 0.55 & 121.5 \\\\
> > \mathrm{Stand}   & 0.031 & 0.48 & 118.2 \\\\
> > \mathrm{Walk}    & 0.021 & 0.55 & 109.8 \\\\
> > \mathrm{Jog}     & 0.034 & 0.64 & 118.5 \\\\
> > \mathrm{Dance}   & 0.041 & 0.48 & 129.5 \\\\
> > \\hline
> > \\end{array}$$

---

> ### Comment · Reviewer_kyb6 · 2024-08-30
>
> Thanks for the responses. Most of my concerns are addressed. I have updated my rating.

---

### Official Review · Reviewer_6vHW · 2024-07-24
**Muscles in Time: Learning to Unfold Embodied Muscular Signals into Motion Sequences**

**Rating:** 5
**Confidence:** 4

**Review:**

This paper presents a synthetic dataset, Muscles in Time (MinT), for simulating muscle activations during human motion using biomechanical models in OpenSim. While the dataset aims to address the limitations of real-world EMG data collection, the reliance on synthetic data raises significant concerns about its practical applicability and generalization to real-world scenarios.

**Strengths:**

- Novel Dataset: The creation of a synthetic muscle activation dataset is a significant contribution that addresses the limitations of real-world data collection.
- Detailed Pipeline: The paper provides a comprehensive description of the pipeline for generating the dataset, which can be useful for future research.
- Robust Experiments: The experiments are well-designed and demonstrate the utility of the dataset for training neural network models.

**Additional Feedback:**

- Integration with Real Data: Exploring the integration of synthetic and real-world data could enhance the robustness and applicability of the dataset.
- Long-Term Implications: Discussing the long-term implications of using synthetic data in biomechanics and potential applications in rehabilitation and sports science could strengthen the paper.

**Clarity:**

While the paper is well-organized and easy to understand, the overemphasis on synthetic data without sufficient real-world validation detracts from its clarity and relevance.

**Correctness:**

The claims made in the submission are theoretically sound but lack practical validation against real-world data, which limits their overall correctness and applicability.

**Documentation:**

The paper includes sufficient detail on the synthetic data collection and preprocessing but lacks comprehensive documentation on how the synthetic data compares to real-world EMG data.

**Ethics:**

There are no significant ethical concerns with the submission. The study adheres to ethical guidelines for research involving human subjects and data privacy.

**Limitations:**

The authors have acknowledged several limitations, including the synthetic nature of the data and the challenges of generalizing to diverse motions. However, the reliance on synthetic data without adequate real-world validation is a significant drawback. Suggestions for improvement include:

- Incorporating real-world EMG data to validate and enhance the synthetic dataset.
- Expanding the dataset to cover a broader range of motions.
- Exploring domain adaptation techniques to mitigate the synthetic-to-real gap.

**Opportunities For Improvement:**

- Real-World Validation: Incorporate real-world EMG data to validate the synthetic dataset and improve its applicability.
- Broader Motion Coverage: Expand the dataset to include a wider variety of motions to enhance generalization.
- Domain Adaptation Techniques: Explore techniques to bridge the gap between synthetic and real-world data.

**Relation To Prior Work:**

The authors provide a thorough review of prior work but fail to adequately address how their approach compares to real-world datasets and methodologies.

**Summary And Contributions:**

This paper introduces the Muscles in Time (MinT) dataset, which simulates muscle activations for various human motions using biomechanical models. The key contributions include:

- Synthetic Dataset Creation: Enrichment of motion capture datasets with simulated muscle activation data using OpenSim.
- Comprehensive Coverage: The dataset covers 227 subjects, 402 muscle strands, and includes over nine hours of simulation data.
- Benchmarking: Evaluation of neural network models for predicting muscle activations from human pose sequences.
- Pipeline Description: Detailed description of the pipeline for integrating motion capture data with biomechanical models to simulate muscle activations.

---

> ### Author Rebuttal · Authors · 2024-08-26
>
> Thank you for your thoughtful feedback and suggestions for improvement. We appreciate your emphasis on the importance of comparing synthetic data to real-world EMG recordings, a concern we take seriously.
>
> As noted in our global rebuttal, directly testing MinT models on real-world data like MiA (Muscles in Action) presents challenges due to issues such as inaccurate pose estimation and subject-specific EMG recording noise. However, we have made efforts to allow for a better comparison of the datasets.
>
> ### **Animated EMG and Motion Graphs**
>
> Regarding your concern about ***"incorporating real-world EMG data to validate the synthetic dataset and improve its applicability,"*** we acknowledge the importance of verifying the usefulness of our dataset. MiA is the most directly comparable real-world dataset to MinT. To facilitate a meaningful comparison, we have provided extensive animated motion sets with corresponding EMG values for both datasets. These animations allow for an in-depth qualitative analysis of EMG pattern similarities and the challenges of generalizing models trained on MinT. You can access the animations for MiA at [https://s.kit.edu/mia-vis](https://s.kit.edu/mia-vis) and for MinT at [https://s.kit.edu/mint-vis](https://s.kit.edu/mint-vis) (tar-files containing videos).
>
> ### **Comparison and Domain Adaptation**
>
> You mentioned the need to ***"explore techniques to bridge the gap between synthetic and real-world data."*** We have created a detailed comparison of selected activity categories, available at [https://s.kit.edu/mint-mia-comparison](https://s.kit.edu/mint-mia-comparison) (video). To facilitate interpretation of the EMG patterns, we marked the lowest compression points of each leg using blue and orange dotted vertical lines for the left and right legs, respectively. Visibly, MiA exhibits significant noise and subject-specific scaling variations, while MinT demonstrates similar but more controlled and consistent patterns. This difference underscores the complexity of directly applying a model trained on MinT to MiA data, as MiA contains strong subject-specific characteristics. MiA itself does not provide a cross-subject split but rather allows a model to learn subject-specific characteristics and exposes the model to the same subjects at test time. While this might be required to achieve acceptable performance at test-time, we believe that MiA would benefit from a cross-subject split to demonstrate the domain gaps between individuals. Likewise, we believe that advanced domain generalization or adaptation techniques are necessary to achieve high performance with little or no training on the MiA target domain dataset when trained on other data like MinT. While this is an important area for future research, it extends beyond the scope of our current paper. We refer to the MiA fine-tuning experiment in Table A5 of the supplementary materials for additional context.
>
> ### **Motion Diversity and Dataset Scope**
>
> We value your suggestion regarding ***"expanding the dataset to cover a broader range of motions."*** since it highlights the importance of a diverse motions set for training. Our dataset, as shown in the animations, offers a broad range of motion classes and significant intra-class diversity in motion execution, distinguishing it from the more controlled MiA dataset motions, which are limited to a much smaller number of different activities. To our knowledge, MinT is the only dataset that combines such a diverse range of motions with detailed muscle activation data, making it uniquely valuable for a wide range of applications.
>
> ### **Limitations and Long-Term Implications**
>
> We recognize the importance of addressing the ***limitations and the need for more discussion on synthetic data"*** and will include a more detailed discussion in the paper. This will include comparisons between the motion patterns observed in MiA and MinT to better contextualize the strengths and weaknesses of our approach.
>
> Regarding your suggestion of ***"Discussing the long-term implications of using synthetic data in biomechanics and potential applications in rehabilitation and sports science [...]"***, we believe that the MinT dataset has significant potential to drive future research in muscle activation-focused machine learning. Specifically, we see MinT as a valuable pre-training dataset that can help models learn the key characteristics of muscle activations induced by motion.
>
> For downstream tasks involving unfiltered raw EMG recordings, advanced generalization techniques—such as introducing noise during training, possibly generated by a model that mimics realistic EMG noise—could be employed. Alternatively, pre-trained models could be fine-tuned using real-world training data specific to smaller-scale tasks. While MinT may not fully replace real-world data in these scenarios, we believe it can significantly reduce the amount of costly real-world data needed, making previously unviable research directions more feasible.
>
> Moreover, for tasks that do not involve raw EMG signals—such as providing semantic embeddings of motion-coupled muscle activations for interpretation by large language models—such generalization techniques are not necessary. This broadens the potential applications of MinT and highlights its versatility in advancing research across different domains.

---

### Official Review · Reviewer_pwPd · 2024-07-24
**Review of the paper: "Muscles in Time: Learning to Understand Human Motion In-Depth by Simulating Muscle Activations"**

**Rating:** 9
**Confidence:** 4

**Review:**

The paper presents a high-quality, original, and significant contribution to the field of human motion analysis and AI. The clarity of the presentation is good, and the work is well introduced and motivated. This work will be helpful for the human motion analysis community, it also offers a very interesting and novel use case for assessing sequence-to-sequence approaches.

**Strengths:**

- The MinT dataset provides a consequence amount of data for muscle activation, covering a wide range of subjects and muscle strands, which is larger than many existing datasets.
- The work allows to bridge the gap between surface-level motion data and internal biomechanical processes. Moreover, as synthetic data is used, it also overcomes the limitation of real EMG data which may be invasive and costly to deploy.
- The paper provides a thorough explanation of the data generation process, including the use of the OpenSim platform and the mapping of virtual markers to the SMPL-H body mesh.
- The authors also demonstrate and assess how the proposed dataset can be used for muscle activation estimation from  human motion sequences.

**Additional Feedback:**

Some small additional comments:
- Table 1 is not referred in the text. I think it would be interesting to discuss more in detail this table to highlights the differences between the proposed dataset and existing ones.
- Between rows 196 and 199, maybe authors can just specify that d=263 for clarity

**Clarity:**

The clarity of the work is very good, with a well motivated and introduced contribution, detailed explanations and a clear methodology.

**Correctness:**

The claims made in the work are largely correct and supported by the evidence provided in the paper.

**Documentation:**

Yes sufficient detail is provided with a clear documentation

**Ethics:**

No ethical concern

**Limitations:**

The authors have acknowledged and addressed several limitations of their work including reliance on simulated data, dataset scope and required manual adjustments. These limitations are clearly discussed.

**Opportunities For Improvement:**

- The authors sometimes provide information about related work or used techniques by merely citing names and references, without offering adequate detail. It hence assumes that the reader is either an expert in the field, already familiar with all the components, or willing to look up each reference while reading.
- Even if authors propose clear documentation about required manual adjustments in the simulation pipeline, it can introduce variability and may slightly affect the reproducibility of the results.

**Relation To Prior Work:**

The paper discussed how this work differs from previous publications and existing datasets. As it is the main contribution, more details could be provided about Table 1.

**Summary And Contributions:**

In this paper, authors introduce a new large-scale synthetic muscle activation dataset called Muscles in Time (MinT). This dataset extends existing motion capture datasets with muscle activation simulations using biomechanical human body models via the OpenSim platform. MinT contains human motion sequences performed by 227 subjects and with 402 simulated muscle strands. Moreover, the authors demonstrate the utility of this dataset by using neural network-based muscle activation estimation from human motion sequences with two different sequence-to-sequence architectures.

---

> ### Author Rebuttal · Authors · 2024-08-26
>
> Thank you for your positive feedback and for recognizing the contributions of our work. We appreciate your suggestions for improvement and are committed to making the necessary enhancements.
>
> ### **Providing More Context in Related Work**
>
> You mentioned that ***"the authors sometimes provide information about related work or used techniques by merely citing names and references, without offering adequate detail."*** We acknowledge this and agree that providing more context can help readers who are not already familiar with the field. In the final paper, we will expand on the discussion of related work and techniques, offering more detailed explanations rather than assuming extensive prior knowledge.
>
> ### **Reproducibility and Manual Adjustments**
>
> We understand your concern regarding the potential variability introduced by the manual adjustments in the simulation pipeline. We recognize that this may impact reproducibility to some extent. To address this, we will publish the exact settings which were used to create this dataset, allowing others to recreate it themselves. The reason why we publish the precomputed results, apart from accessibility, is that performing the optimization with OpenSimAD is highly compute intensive. While everyone can use our settings to recreate the calculations for individual samples, recreating the whole dataset comes with large computing cost.
>
> ### **Discussion of Table 1**
>
> You noted that *"Table 1 is not referred in the text"* and suggested a more detailed discussion to highlight the differences between the proposed dataset and existing ones. We agree that this would be valuable. In the final paper, we will explicitly refer to Table 1 in the text and provide a more detailed comparison of MinT with other datasets, explaining the specific contributions of our dataset.
>
> ### **Clarification in the Methodology**
>
> Regarding your suggestion between rows 196 and 199 to *"specify that d=263 for clarity,"* we appreciate the attention to detail and will make this change to improve readability.
>
>
>
> We are grateful for your thorough review and positive evaluation and are committed to addressing these points in the final paper.

---

### Official Review · Reviewer_Bfvp · 2024-07-25
**Simulated large new muscles and motion dataset**

**Rating:** 8
**Confidence:** 4
**Clarity:** The paper is written very clearly.

**Review:**

The paper is written clearly and details how the data was compiled and measures taken to reduce the errors. The paper also details assumptions made so that the authors understand the limitations. Future researchers can use both the method and the data for future work. Overall, it would be a valuable dataset for researchers.

**Strengths:**

- Compiled new large dataset
- Comprehensive pipeline/ code to convert motion data into muscle activation
- Evaluation of the data through benchmarks and choice of metrics

**Additional Feedback:**

N/A

**Correctness:**

Yes, they are correct. However, there is no way to know the ground truth without making actual muscles measurements. For such a large dataset, this is a good milestone work which was validated using established bio-mechanical models.

**Documentation:**

No, the documentation of the dataset is severely lacking. The authors are advised to expand their documentation so that future researchers can easily follow instructions to reproduce the results and apply the published techniques for their research.

**Ethics:**

No, I do not.

**Limitations:**

There are significant assumptions made for the work to be possible. This paves a milestone for having a perfect future model. Authors did a decent job of documenting many limitations and assumptions, however, there are still more that are missing. They have been mentioned in the room for improvements section. They are advised to make the paper even better by including the aforementioned suggestions.

**Opportunities For Improvement:**

- SMPL is known to not represent well feet and hands. It is not clear in the paper, how did you compensate or overcome this challenge?
- Line 119-122: This is clearly a major limitation. This raises major questions on what the ground truth is and if this kind of dataset will lead future researchers to assume an average model can satisfy the requirements for understanding body kinematics which could also have severe health care repercussions.
- Line 25: The 67 strategically placed joints is not clearly marked in the figure. Please elaborate and define each marker location and how the markers need to be placed so future researcher and reproduce your work and apply towards their research using work reported in this paper.
- In supplemental work (Line 597-605), the code https://github.com/simplexsigil/MusclesInTime is listed. However, the repo lack good readme file, nor does the supplemental file have good instructions to use the data and code. Please elaborate so research community can benefit from your work. Publishing dataset and pipeline without proper documentation is pointless as no one will be able to use it.
- Since it is impossible to have ground truth in a simulated dataset, esp with many assumptions, it is recommended that the authors using existing smaller datasets to validate their learned model on real datasets.

**Relation To Prior Work:**

This is the first paper to the best of my knowledge that simulated such a large muscle activation during various activities with so many different subjects.

**Summary And Contributions:**

This work combines many existing datasets, pipelines, algorithms, and optimizes some to finally produce a new large muscles activation dataset. It consists of large number of subjects with varying background and body measurements. Finally they demonstrate the usefulness of the dataset by predicting muscle activity during various activities with 3 different metrics.

---

> ### Comment · Reviewer_Bfvp · 2024-08-17
>
> Thanks for the reply. I thank the author in adding more detailed instructions on the repository for future researchers. After reading other reviewer's evaluation and author's rebuttal, I stand by my review score.

---

### Author Rebuttal · Authors · 2024-08-16

Dear reviewers,
dear chairs,
we are thankful and appreciate the generally positive feedback on our work as well as the constructive comments and suggestions. We address common key questions and suggestions from the reviewers \bfvp, \pwpd, \svhw, and \kybs in the provided PDF file, we will further address the reviewers directly with more detailed responses.

---

### Decision · Program_Chairs · 2024-09-26

**Decision:**

Accept (Poster)

**Comment:**

This paper presents a large synthetic muscle activation dataset. The contribution is to augment a set of existing recordings with simulated data based on a physical model. Overall, I believe that the reviewers consensus supports accepting this paper.   I would advise that the authors double check that they have added as much documentation to the dataset as possible to enable it to be as useful as possible.  I would also suggest they consider expanding on the limitations - perhaps in an appendix. Simulated data often have a number of limitations and these can be problematic for people adopting them who are unaware of the full set of assumptions that were made in creating them.